# Pathway choice in the alternative telomere lengthening in neoplasia is dictated by replication fork processing mediated by EXD2's nuclease activity

Ronan Broderick[1], Veronica Cherdyntseva[2], Jadwiga Nieminuszcy [1], Eleni Dragona[2], Maria Kyriakaki[2], Theodora Evmorfopoulou[2], Sarantis Gagos [2,3] ✉ & Wojciech Niedzwiedz [1,3] ✉

Telomerase-independent cancer proliferation via the alternative lengthening of telomeres (ALT) relies upon two distinct, largely uncharacterized, break-induced-replication (BIR) processes. How cancer cells initiate and regulate these terminal repair mechanisms is unknown. Here, we establish that the EXD2 nuclease is recruited to ALT telomeres to direct their maintenance. We demonstrate that EXD2 loss leads to telomere shortening, elevated telomeric sister chromatid exchanges, C-circle formation as well as BIR-mediated telomeric replication. We discover that EXD2 fork-processing activity triggers a switch between RAD52-dependent and -independent ALT-associated BIR. The latter is suppressed by EXD2 but depends specifically on the fork remodeler SMARCAL1 and the MUS81 nuclease. Thus, our findings suggest that processing of stalled replication forks orchestrates elongation pathway choice at ALT telomeres. Finally, we show that co-depletion of EXD2 with BLM, DNA2 or POLD3 confers synthetic lethality in ALT cells, identifying EXD2 as a potential druggable target for ALT-reliant cancers.

To achieve unlimited proliferation potential, cells must maintain their telomeres. However, due to the end replication problem, telomeres are progressively depleted after consequent cell divisions[1]. When critically short telomeres occur, DNA damage responses (DDR) are activated, and cells undergo growth arrest[2,3]. To sustain continuous cell growth, most cancers activate the reverse transcriptase telomerase[4,5]. However, 10–15% of cancers, especially tumours of mesenchymal origin (i.e., 30–50% of osteosarcomas, soft tissue sarcomas or primary brain tumours) support cell proliferation via the Alternative Lengthening of Telomeres (ALT)[6]. These represent aggressive malignancies, and presently, there are limited therapeutic options for their treatment.

Cancer cells utilizing ALT display telomere length heterogeneity and elevated chromosomal instability (CIN)[7,8]. ALT-positive (ALT+) cells are characterised by increased incidence of extrachromosomal circular telomeric DNA (C-circles) and an elevated frequency of telomeric sister chromatid exchanges (T-SCEs)[7]. ALT is predominantly carried out at ALT-associated PML bodies (APBs), which contain homology-directed DDR factors such as RAD51, RAD52, BLM and the MRN complex where clustered telomere ends undergo recombinatorial ALT activity[7,9].

Seminal work in survivors of telomerase null yeast mutants, identified two main pathways by which ALT is operated, both requiring *RAD52*, indicating homologous recombination (HR)-driven

---

[1]The Institute of Cancer Research, London, UK. [2]Laboratory of Genetics, Center of Clinical Research, Experimental Surgery and Translational Research Biomedical Research Foundation Academy of Athens (BRFAA), Athens, Greece. [3]These authors contributed equally: Sarantis Gagos, Wojciech Niedzwiedz. ✉e-mail: sgagos@bioacademy.gr; wojciech.niedzwiedz@icr.ac.uk

processes[10,11]. Furthermore, *RAD50* and *RAD51* were shown to control yeast survival independently of one another[12], with *RAD51*, *RAD54* and *RAD57*[12] mediating Type I survivors. In contrast, Type II survivors rely on *RAD50* and the *XRS2/MRE11* (yeast homologues of the human MRN complex)[12] and *RAD59*[12]. Importantly, ALT in yeast was proposed to operate via Break-Induced Replication (BIR) mediated by Pol32[11], which may rely on conservative DNA synthesis[13]. Recent work carried out by the Malkova laboratory suggests that both Type I and Type II mechanisms may operate simultaneously to promote a "unified ALT survivor pathway", at least in yeast[14]. However, Type I and Type II mechanisms may not be exactly analogous to RAD52-dependent and independent ALT in humans.

The description of an analogous BIR mechanism underlying ALT in human neoplasia, thought to be initiated by collapsed replication forks (RFs)[15,16] was shown to be dependent on RAD52-mediated strand annealing and the yeast pol32 homologue POLD3/4[9,16] with subsequent studies identifying a RAD52-independent mechanism that was also implicated in ALT-dependent telomere maintenance[17,18].

While both ALT pathways rely on BLM and PCNA-RFC-Polδ, the RAD52-dependent pathway is considered the dominant mechanism of telomere lengthening[19] whereas the RAD52-independent is associated with elevated production of the ALT-characteristic C-circles[16,18]. Both types of ALT-associated BIR are expected to proceed via conservative telomere neosynthesis[13,16,20,21]. Interestingly, the RAD52-independent ALT pathway is purported to maintain telomeres independently of RAD51 and MRE11, while the ablation of these HR factors is associated with C-circle overproduction[18]. However, how DNA repair via these two pathways is triggered to drive ALT-mediated telomere elongation, as well as their components and mechanisms remain enigmatic.

Here, we provide evidence that replication fork processing by EXD2, a 3′–5′ exonuclease[22,23], plays an important role in productive telomere synthesis. By doing so, EXD2 determines repair pathway choice within the ALT mechanism, suppressing RAD52-independent BIR. Accordingly, EXD2 localises to telomeres and ALT-associated PML bodies (APBs) and is expressed in a panel of ALT-reliant cell lines. Despite the concomitant impaired telomere maintenance, loss of EXD2 results in hyper-ALT phenotypes including increased C-circle accumulation and elevated frequencies of T-SCEs and APBs. Mechanistically, we show that pathological processing of stalled replication forks in ALT cells lacking EXD2 diverts their telomere maintenance from RAD52-mediated recombination, towards RAD52-independent BIR. Additionally, we reveal that this latter form of telomere length restoration is associated with elevated frequencies of conservative replication and relies on fork regression mediated by the SMARCAL1 remodeler and the MUS81 structure-specific nuclease.

Together, our findings establish that initial nucleolytic processing of the stalled replisome orchestrates repair pathway choice for ALT telomere maintenance. Moreover, we also uncover that loss of EXD2 is synthetic sick with BLM, DNA2 and POLD3, which could represent an attractive therapeutic target against ALT-dependent cancers.

## Results

### EXD2 is recruited to ALT telomeres and promotes their maintenance

The EXD2 nuclease plays a critical role in promoting both, homology directed repair as well as stability of dysfunctional replication forks[22,23]—two processes that underpin ALT, making it a likely candidate ALT-mediator. We tested this hypothesis in several ways. First, we analysed if EXD2 localises to telomeres in ALT-reliant U2OS cells stably expressing GFP-EXD2 to near WT levels (Supplementary Fig 1a). We could readily detect significant co-localisation of an antibody specific to the shelterin protein TRF2 with GFP-EXD2 (Fig. 1a, quantified in b) as well as co-localisation of GFP-EXD2 to ALT-associated PML bodies, which are a characteristic feature of ALT-reliant cells[7] by four-colour immunofluorescence analysis (Fig. 1a, quantified in c). Next, we

employed the proximity ligation assay (PLA) using antibodies against GFP- or FLAG-tagged EXD2 and TRF2 (Fig. 1d, e). Again, the PLA signal between tagged-EXD2 and TRF2 suggests that these proteins are proximal to one another. Taken together these data suggest a putative role for EXD2 within the ALT mechanism.

Given EXD2's localization to ALT telomeres and APBs, we tested if loss of EXD2 induces telomere dysfunction[24]. To this end, we analysed the frequency of Telomere Dysfunction-Induced Foci (TIF) formation at telomeres utilising Immuno-FISH staining for 53BP1 (a DSB marker) and a telomere-specific FISH probe. We observed a significant increase in TIFs in the absence of EXD2 compared to control U2OS cells (Fig. 2a, Supplementary Fig. 1b). These results were recapitulated employing the proximity ligation assay (PLA) using antibodies recognising TRF1 or TRF2 and 53BP1 and by immunofluorescence staining for γH2AX (an independent DSB marker) and TRF2 (Fig. 2b, Supplementary Fig. 1c, d), as well as in IMR90 cells (an independent ALT-reliant cell line) depleted for EXD2 (Supplementary Fig. 1e). Importantly, this increase in telomere dysfunction is suppressed by the nuclease activity of EXD2, as only the WT EXD2 but not the nuclease-dead mutant version of the protein[22,23] rescues this phenotype (Fig. 2c, Supplementary Fig. 1f).

To address the origin of increased terminal breakage, we analysed the stability of replication forks duplicating telomeric DNA. EXD2-deficient cells displayed elevated levels of global replication fork asymmetry—a marker of increased fork stalling (Supplementary Fig. 1g). Furthermore, dual-colour telomere-strand specific FISH, revealed an increased incidence of unreplicated telomeres, as indicated by loss of telomeric fluorescence in one of the sister chromatids in EXD2-deficient cells compared to parental control U2OS cells (Fig. 2d). Finally, EXD2 loss led to increased telomeric RPA localisation at telomeres, again suggesting increased rates of terminal fork collapse/processing (Fig. 2e). Collectively, these phenotypes are indicative of accelerated DNA damage responses (DDR) at the ALT telomere likely resulting from collapsed replication forks upon EXD2 loss.

To further shed light on the mechanism associated with the formation of TIF in EXD2-deficient cells, we addressed the contribution of early molecular events (fork reversal) *vs* late (fork resection/recombination). First, we analysed the effect of HR ablation, by chemical inhibition of MRE11 or RAD51 on the overall load of TIF in *EXD2*−/− cells. We observed a significant TIF increase in WT cells upon Mirin or RAD51-inhibitor (RAD51i) treatment but importantly, there was no further increase in dysfunctional telomeres in EXD2-deficient cells, suggesting that EXD2-dependent telomeric fork processing is epistatic with the HR machinery, likely providing the substrate required for the initiation of HR-dependent fork restart (Fig. 2f, Supplementary Fig. 1h). Secondly, we took advantage of the fact that EXD2 counteracts SMARCAL1 in regulating fork regression[23] and that SMARCAL1 is dispensable for classical HR[25]. Strikingly, SMARCAL1 knockdown by two independent siRNAs resulted in almost complete rescue of the excess TIF observed in *EXD2*−/− U2OS as compared to WT (Fig. 2g, Supplementary Fig. 1i, j). This provides evidence that EXD2-dependent replication fork processing promotes DNA synthesis at telomeres, suppresses telomeric DNA breaks and regulates initiation of various types of homology-mediated terminal repair[26].

### EXD2 loss leads to hyperactivation of ALT-associated phenotypes and telomere shortening

Given the above, we next tested if EXD2 loss is associated with the modulation of ALT-characteristic phenotypes. We observed an increase in extrachromosomal C-circle DNA in EXD2-deficient U2OS cells, (importantly this was also recapitulated in a panel of ALT-positive cells depleted for EXD2 by siRNA) (Fig. 3a, b, Supplementary Fig. 2a). Interestingly, increased C-circle levels upon EXD2 depletion were not observed in a derivative of the ALT + VA-13 cell line that re-expresses telomerase and represses ALT (VA-13+hTel)[27] suggesting that the excess C-circles in EXD2 depleted cells, are ALT-specific, and may

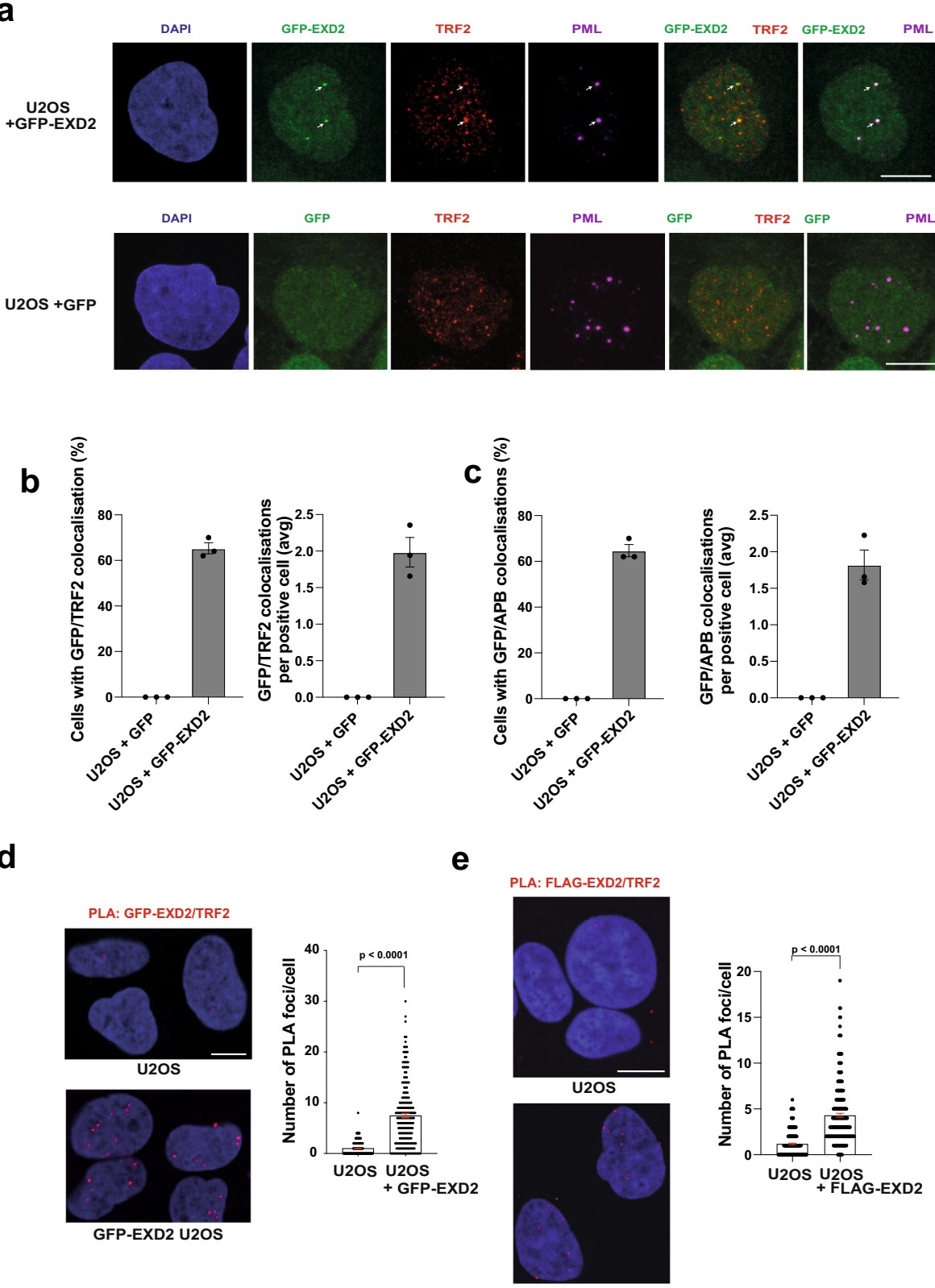

represent hyperactivation of the RAD52 independent telomere lengthening in the absence of telomerase (Fig. 3b). Accordingly, EXD2 was ubiquitously expressed in a panel of ALT-reliant cell lines and interestingly, the VA-13+hTel line showed a marked decrease in EXD2 protein levels compared to the parental ALT+, VA-13 (Fig. 3c), suggesting a possible correlation between EXD2 expression and ALT

efficiency. In line with a role for EXD2 in ALT, its absence resulted in a significant increase in ALT-associated PML bodies in both EXD2-deficient U2OS and IMR90 cells depleted for EXD2 by siRNA treatment (Fig. 3d, Supplementary Fig. 2a, b) as well as increased levels of T-SCEs, both ALT hallmark phenotypes, compared to parental control cells (Fig. 3e). This likely indicates either, an increased use of the ALT-

**Fig. 1 | EXD2 is recruited to dysfunctional telomeres and co-localises with ALT-associated PML bodies. a** GFP-EXD2 (green) localises to ALT-associated PML bodies as determined by 4-colour immunofluorescence staining carried out in control U2OS cells expressing GFP only and U2OS cells stably expressing GFP-EXD2 using antibodies raised against TRF2 (red) and PML (magenta). DAPI (blue) acts as a nuclear stain ($n = 3$ independent experiments, scale bar = 10 μm). **b** Quantification of GFP/TRF2 co-localisations (percentage cells with co-localisations and average number of co-localisations per positive cell) in cells from (**a**), $n = 150$ cells examined over 3 independent experiments, bars represent +/− SEM. **c** Quantification of GFP/ APB co-localisations (percentage cells with co-localisations and average number of co-localisations per positive cell; in cells from **a**), $n = 150$ cells examined over 3 independent experiments, bars represent +/− SEM. **d, e** GFP-EXD2 and FLAG-EXD2 localise to telomeres as assayed by its association with TRF2 (Shelterin component) by the PLA assay. The number of PLA foci per cell is quantified in U2OS control cells or cells stably expressing GFP- or FLAG-EXD2. PLA signal appears in red, DAPI (blue) acts as a nuclear stain ($n = 194$ vs. 204 cells (**d**) and 211 vs. 258 cells (**e**) examined over 3 independent experiments, statistical analysis was carried out by two-sided Mann–Whitney test, error bars represent +/− SEM, scale bar = 10 μm). Source data are provided as a source data file.

mechanism in these cells or a switch to an alternative telomere repair pathway within the ALT-mechanism, which drives the increased incidence or persistence of APBs. Interestingly, *EXD2*−/− U2OS cells displayed a significant decrease in average telomere length, as analysed by a modified metaphase Q-FISH protocol (Fig. 3f) and DNA combing as carried out previously[28] (Fig. 3g), highlighting the impact of EXD2 loss on telomere maintenance in ALT cells.

### EXD2 regulates repair pathway choice at ALT telomeres

Given the hyperactivation of the ALT phenotypes coupled with poor telomere maintenance observed in EXD2-deficient ALT-reliant cells, we hypothesised that these cells may employ an inefficient/aberrant telomere maintenance mechanism hampering telomerase-independent telomere elongation. Thus, we sought to establish the repair mechanism(s) by which EXD2-associated T-DSBs are mended. Given the epistatic relationship between EXD2 and RAD51 or MRE11 (Fig. 2f, Supplementary Fig. 1h) in the repair of telomeric-DSBs and the fact that EXD2 is also required for Alt-EJ[23] and Supplementary Fig. 2c, we rule these out as the main repair pathways utilised by ALT cells in the absence of EXD2.

To establish the impact of EXD2 loss on the two known forms of ALT-associated BIR, we examined the frequency of EdU incorporation at ALT telomeres by immuno-FISH in cells synchronised in the G2 phase of the cell cycle[18] in presence or absence of siRNA targeting RAD52 (schematic diagrams, Fig. 4a, Supplementary Fig. 3a). Strikingly, we observed that loss of EXD2 resulted in highly elevated levels of telomeric EdU incorporation compared to control cells. This incorporation was not affected by RAD52 depletion in *EXD2*−/− cells, but was markedly reduced in the controls, indicating that EXD2 loss diverts repair towards RAD52-independent terminal DNA synthesis (Fig. 4a).

We then examined the frequency of BIR at telomeric sites in synchronised prometaphase cells as previously described[29] (schematic diagram, Fig. 4b). Loss of EXD2 resulted in elevated levels of telomeric EdU incorporation in prometaphase cells compared to WT controls, again indicating an increased incidence of ALT-associated BIR. In agreement with our data generated using siRNA targeting RAD52 in G2-synchronised cells, we also observed reduced frequency of EdU incorporation in WT, but not in *EXD2*−/− cells upon presence of a RAD52 inhibitor (AICAR) (Fig. 4b). Importantly, this phenotype was also recapitulated in IMR90, another ALT-reliant cell line (Supplementary Fig. 3b). Moreover, the nuclease activity of EXD2 is essential to suppress elevated EdU incorporation at telomeres (Supplementary Fig. 3c). Interestingly, inhibition of RAD51 alone or concomitant inhibition of both RAD51 and RAD52 in EXD2-deficient U2OS cells fails to suppress this excess telomeric EdU incorporation in isolated synchronised prometaphase cells, suggesting that RAD51 is not required to mediate telomere synthesis in prometaphase in the absence of EXD2 (Supplementary Fig. 3d). This function of EXD2 seems epistatic with canonical HR, as treatment with the MRE11 inhibitor Mirin increased frequency of EdU incorporation in control prometaphase cells (BIR) but had no effect in *EXD2*−/− cells (Supplementary Fig. 4a). Moreover, *EXD2*−/− cells treated with siRNA targeting MRE11 or RAD51 showed similarly elevated levels of C-circles, (Supplementary Fig. 4b, c). Since RAD52-independent BIR is associated with increased incidence of

extrachromosomal C-circle DNA in the absence of MRE11 or RAD51[18] our results demonstrate that EXD2-dependent fork processing suppresses RAD52-independent BIR.

### Telomere maintenance via RAD52-independent BIR is associated with high frequencies of conservative replication

Our analysis above indicated that EXD2 may function as a molecular switch regulating the two branches of ALT-associated BIR. To further investigate our observations, we utilised a modification of the segregated CO-FISH technique[30], capable to determine the frequency of BIR-mediated conservative telomeric DNA synthesis by following the segregation of template telomeric DNA strands between sister chromatids for two consequent replication rounds (Supplementary Fig. 5a, b, scenarios viii and ix of panel b were scored as conservative telomere synthesis events). Strikingly, this analysis showed that EXD2 deficiency results in increased frequencies of BIR-mediated conservative telomeric replication, likely to reflect recombinatorial interactions between sister chromatids (Fig. 4c). Similar results were obtained when we compared control U2OS cells stably transfected to conditionally overexpress the Cyclin-E replication factor, with two clones of analogous isogenic cells additionally rendered RAD52 knockout by CRISPR/Cas9[31]. These findings support the notion that EXD2 loss pushes cells towards the use of RAD52-independent conservative BIR (Fig. 4c). To gain further insight into the regulation of the two distinct ALT-associated BIR mechanisms, we took advantage of the fact that HR-like vs BIR-like repair events can be differentiated by examining EdU incorporation on metaphase chromosomes, to distinguish between semi-conservative (HR-mediated) and conservative (BIR-mediated) fork restart events[13,21]. This analysis confirmed that EXD2-deficient cells employed more often the conservative (BIR) form of DNA synthesis at their telomeres compared to WT controls (schematic diagram, Supplementary Fig. 5c). In further support of the notion that loss of EXD2 pushes cells towards engagement of RAD52-independent BIR, we also observed a decreased association of RAD52 to telomeres in EXD2-deficient U2OS cells compared to WT, while association of RAD51 is unchanged (Supplementary Fig. 6a, b). Altogether, these results unravel the initial molecular events regulating the terminal repair pathways at ALT telomeres suggesting a key role for EXD2-dependent fork processing in orchestrating homology-directed telomere maintenance.

### Genetic requirements for RAD52-independent ALT telomere maintenance

Previous work has indicated critical but opposing roles of the BLM helicase and the SLX4 nuclease in supporting efficient ALT by coordinating resolution and dissolution of recombining telomeres[32]. Thus, we sought to establish the genetic relationship between these factors and EXD2 in suppressing RAD52-independent BIR. To this end, we measured the incidence of telomeric EdU incorporation in isolated synchronised prometaphases in U2OS control or *EXD2*−/− cells treated with control siRNA or siRNA targeting either BLM or SLX4. As expected, BLM depletion caused a marked reduction in EdU incorporation in all cell lines[18,33]. SLX4 depletion resulted in no change in the levels of EdU incorporation in *EXD2*−/− cells while

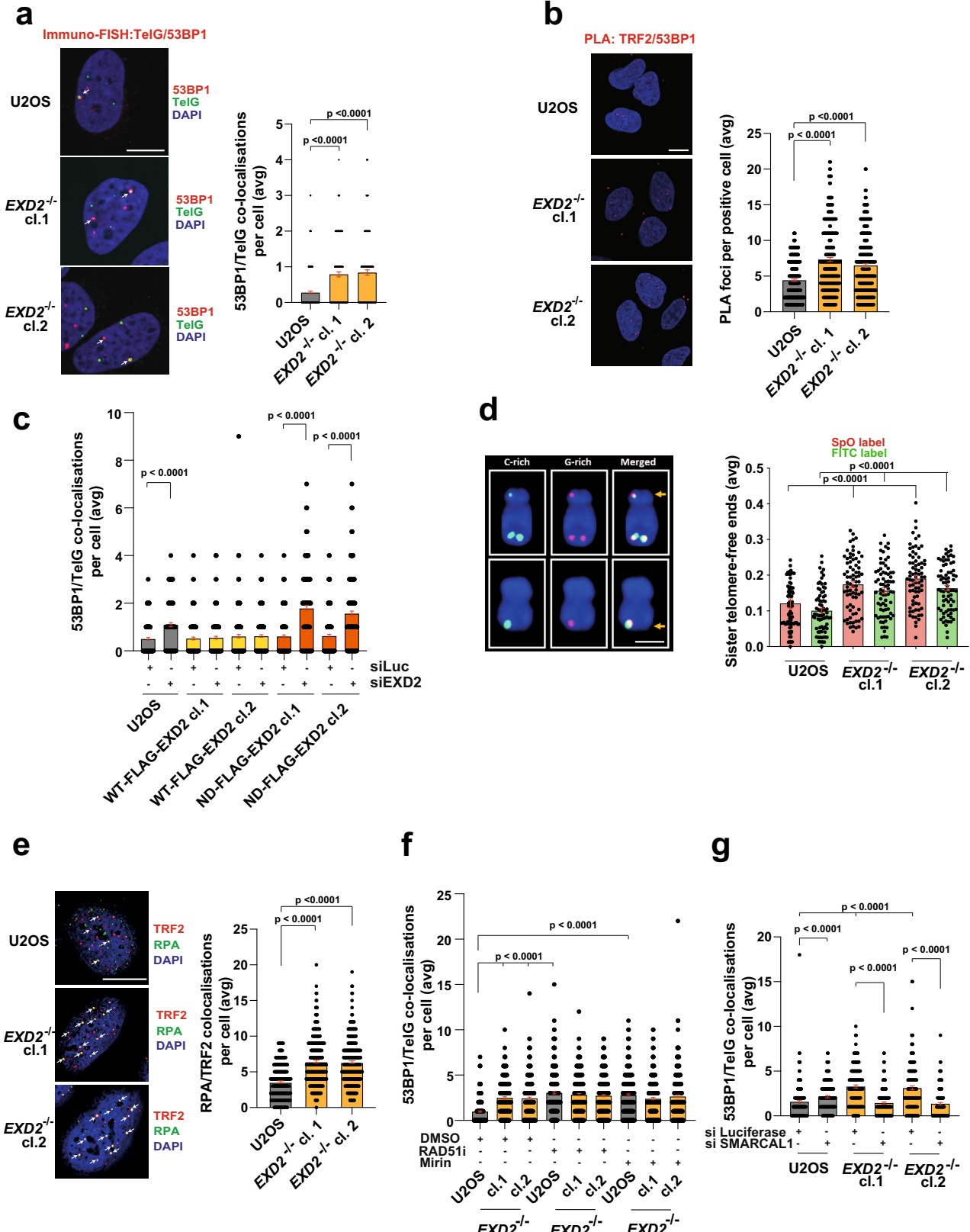

increasing the frequency of events in control cells, suggesting that EXD2 and SLX4 act within the same pathway to suppress RAD52-independent BIR in ALT cells (Fig. 5a, Supplementary Fig. 7a, b). We next tested whether fork regression plays a role in the initiation of ALT-associated RAD52-independent BIR. Indeed, we observed increased association of the fork remodeler SMARCAL1 to telomeres

in EXD2-deficient cells (Supplementary Fig. 7c). Strikingly, depletion of SMARCAL1 by two independent siRNAs reduced the elevated incidence of EdU incorporation upon EXD2 loss to near WT levels suggesting, that unscheduled fork regression is likely a key initial event driving BIR-dependent telomere maintenance (Fig. 5a, Supplementary Fig. 7d).

**Fig. 2 | EXD2-deficient cells display increased telomere dysfunction. a** Immuno-FISH staining in WT and *EXD2*^-/- U2OS cells. 53BP1 (red) acts as a DSB marker, TelG PNA FISH probe staining (green) acts as a marker for the telomere and DAPI (blue) acts as a nuclear stain (*n* = 219, 172 and 175 cells examined over 3 independent experiments, statistical analysis by two-sided Mann–Whitney test, bars represent +/− SEM, scale bar = 10 μm). **b** PLA assay carried out in WT and *EXD2*^-/- U2OS cells using antibodies recognising TRF2 (shelterin component) and 53BP1 (DSB marker). PLA signal appears in red, DAPI (blue) acts as a nuclear stain (*n* = 297, 259 and 281 cells examined over 3 independent experiments, statistical analysis was carried out by two-sided Mann–Whitney test, bars represent +/− SEM, scale bar = 10 μm). **c** Quantification of 53BP1/TelG co-localisations per nucleus by immuno-FISH staining in WT U2OS cells or cells stably overexpressing either WT or nuclease-dead mutant FLAG-HA EXD2, treated with either control siRNA or siRNA targeting the 3′ UTR of endogenous EXD2 (*n* = 168, 152, 165, 190, 218, 223, 203, 181, 203 and 192 cells examined over 2 independent experiments. Statistical significance was determined by two-sided Mann–Whitney test, bars represent +/− SEM). **d** Quantification of sister telomere-free ends using telomere strand-specific PNA FISH probes (SpO and FITC-labelled, red, and green signals, respectively) in U2OS control and *EXD2*^-/- cells

(*n* = 75 metaphases per sample examined over 3 independent experiments, statistical analysis by two-sided Mann–Whitney test, bars represent +/− SEM, scale bar = 1 μm). **e** Quantification of RPA/TRF2 co-localisations (green and red signals, respectively) in U2OS WT and *EXD2*^-/- cells. DAPI (blue) acts as a DNA stain (*n* = 303, 305 and 315 cells examined over 3 independent experiments, statistical analysis by two-sided Mann–Whitney test, bars represent +/− SEM, scale bar = 10 μm). **f** Immuno-FISH analysis of WT and *EXD2*^-/- U2OS cells treated with DMSO or RAD51 inhibitor (B-02, 25 μM, 2 h treatment) or Mirin (50 μM, 1 h treatment). 53BP1 acts as a DSB marker, TelG PNA FISH probe staining acts as a marker for the telomere and DAPI acts as a nuclear stain (*n* = 301, 305, 300, 302, 302, 300, 300, 301 and 302 cells examined over 3 independent experiments, statistical analysis by two-sided Mann–Whitney test, bars represent +/− SEM, scale bar = 10 μm). **g** Quantification of 53BP1 co-localisations with TelG telomeric probe by immuno-FISH staining in WT and *EXD2*^-/- U2OS cells treated with either control siRNA or siRNA targeting SMARCAL1 (as indicated) (*n* = 150 cells examined over 3 independent experiments, statistical analysis was carried out by two-sided Mann–Whitney test). Source data are provided as a source data file.

Since SMARCAL1 depletion rescued the excess EdU incorporation and the frequency of replication fork collapse at the telomere in the absence of EXD2, we hypothesised that SMARCAL1 may also impact conservative telomeric synthesis. Indeed, analysis of conservative telomeric synthesis revealed that this is the case, indicating that nucleolytic processing of reversed forks by EXD2 dictates ALT pathway choice (Fig. 5b). Moreover, given that regressed forks can be cleaved by MUS81 to drive POLD3-dependent synthesis[34] and our observations revealing increased association of MUS81 to telomeres in EXD2-deficient cells (Supplementary Fig. 8a), we hypothesised that fork regression-dependent DNA synthesis may be employed in ALT + cells lacking EXD2. In support of this hypothesis, depletion of MUS81 reduced the incidence of telomeric conservative DNA synthesis observed (likely intra-chromosomal events) in *EXD2*^-/- U2OS to near WT levels as assayed by segregated two-replication round CO-FISH analysis of metaphase chromosomes (Fig. 5b). We observed the same reduction in synthesis when analysing EdU incorporation in prometaphase cells using two independent siRNAs against MUS81 (Supplementary Fig. 7d and Supplementary Fig. 8b). Moreover, the frequency of TIFs in EXD2-deficient cells was also reduced to WT levels by MUS81 depletion using two independent siRNAs (Supplementary Fig. 8c and Supplementary Fig. 1j). Furthermore, MUS81 or SMARCAL1 depletion did not have any significant impact on cell survival of *EXD2*^-/- cells as measured by colony formation assays (Supplementary Figs. 8d and 9f). These data indicate that loss of EXD2 results in stalled forks that act as a substrate for MUS81-dependent cleavage at the telomere, and that such cleavage is required for the engagement of RAD52-independent BIR.

One prediction of our data would be that the shift towards RAD52-independent BIR may lead to either conservative telomeric replication or to telomeric-sister chromatid exchanges (T-SCE), which may arise via nucleolytic cleavage and resolution of recombination intermediates during the repair of one-ended or two-ended telomeric DSB and may involve sister chromatids[32,35]. In support of this idea, the increased T-SCEs observed in EXD2-deficient cells compared to parental control cells was rescued to near WT levels by depletion of either SMARCAL1 or MUS81 (Fig. 5c). Given the impact of EXD2 on cells' ability to utilise BIR, we hypothesised that its loss may confer synthetic sickness with depleted factors promoting this type of DNA repair. In line with this hypothesis, BLM, which mediates both arms of the RAD52-dependent and -independent ALT[18], is required for survival in EXD2-deficient U2OS cells as assayed by two independent siRNAs (Fig. 6a, Supplementary Fig. 9a). Interestingly, SLX4 which was recently shown to have a synthetic lethal relationship with the SLX4IP protein[32,36] is not required for survival of EXD2-deficient U2OS cells (Fig. 6a). Moreover, EXD2 loss is also synthetic sick with depletion of

the DNA2 nuclease (which is recruited by BLM to APBs to promote telomere synthesis[33]) as determined using two independent siRNAs targeting DNA2 (Fig. 6b, Supplementary Fig. 9b, c).

POLD3 is required for both RAD52-depedent and -independent DNA synthesis during BIR at ALT telomeres[18] and in line with the role of EXD2 in regulating ALT-associated BIR, depletion of POLD3 by two independent siRNAs in *EXD2*^-/- cells results in a marked reduction in proliferative capacity compared to control cells (Fig. 6c, Supplementary Fig. 9d, e). We believe that these synthetic-sick relationships are a consequence of EXD2-deficient cells utilising RAD52-independent BIR as the primary means of telomere maintenance, since targeting factors which promote both arms of ALT synthesis confer lethality, while depletion of RAD52 (required for only one ALT arm) and SMARCAL1 (which causes fork regression leading to increased use of RAD52-independent BIR in EXD2-deficient cells) did not confer synthetic sick phenotypes in *EXD2*^-/- cells (Supplementary Fig. 9f). Indeed, we observe increased association to telomeres and APBs of factors known to promote ALT synthesis (i.e., SLX4, MRE11 and POLD3) in EXD2-deficient cells (Supplementary Fig. 10a–c).

## Discussion

Taken together, our analyses reveal the early molecular events that drive repair pathway choice within the ALT mechanism, implicating nucleolytic processing of regressed replication forks as a key molecular switch that orchestrates homology-directed telomere maintenance. Specifically, we show that the initial replication fork processing by EXD2/SMARCAL1 and MUS81 determines the repair mechanism at DNA breaks occurring in ALT telomeres, thus connecting fork processing with the engagement of homology-directed telomeric DNA synthesis via conservative RAD52-independent BIR[17,18]. Moreover, this mechanism is frequently employed by RAD52-deficient ALT cells, since *RAD52*^-/- U2OS cells displayed elevated levels of conservative DNA synthesis as assayed by two-replication round segregated CO-FISH[16].

Mechanistically, we propose that loss of EXD2 would result in excess replication fork reversal at the telomere, mediated by fork remodelers such as SMARCAL1 (itself shown to modulate replication stress at ALT telomeres[26]). Under normal conditions, reversal, and remodelling of stalled telomeric replication forks prevents the formation of MUS81-dependent DNA DSBs[37]. Hence, uncontrolled fork reversal driven by EXD2 loss may lead to nucleolytic cleavage by MUS81 to generate a substrate (i.e., a single-ended DSB and/or a D-loop structure) that can then be efficiently engaged by RAD52-independent BIR. Notably, this appears to be mechanistically distinct to RAD52-dependent BIR, which has been associated with telomere clustering at APBs[9,33] and does not require MUS81 to engage DNA synthesis at

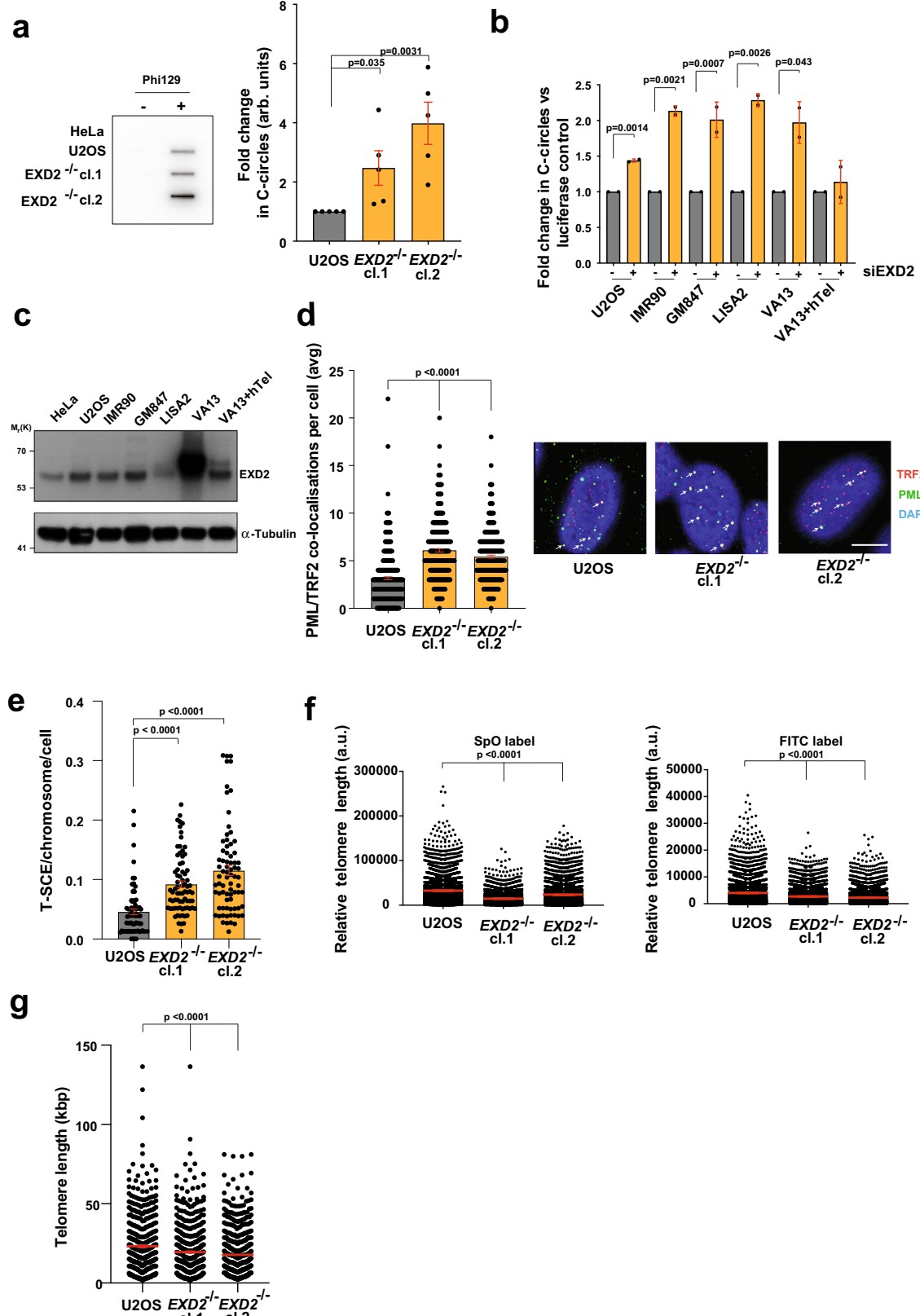

telomeres of ALT-reliant cells[38]. MUS81 is involved in the generation of ALT-associated T-SCEs[39] and has been shown to promote cleavage of regressed degraded forks in BRCA2-deficient cells to allow POLD3-dependent synthesis[34]. Moreover, work in yeast from the Ira lab identified a role for MUS81 in limiting BIR-mediated DNA synthesis by cleavage of migrating D-loops to allow fork rescue by a converging fork[40]. Importantly, at most human telomeres the terminal fork cannot be rescued by a converging fork[41], thus the migrating D-Loop arising from BIR may undergo multiple MUS81-dependent cleavages. This in turn, may result in termination of ongoing D-Loop migration and conservative synthesis or multiple re-engagement events in a single cell cycle[42]. The template utilized could be either a sister telomere

**Fig. 3 | EXD2-deficient cells exhibit hyperactivation of ALT-associated pheno-types and poor telomere maintenance. a** Quantification of extrachromosomal telomeric C-circles by rolling circle PCR amplification and radio-labelled slot blot-ting using a probe complimentary to the telomeric repeat sequences in U2OS control and *EXD2*−/− cells. HeLa samples act as a non-ALT control. Samples pro-cessed without Phi129 polymerase act as a polymerase-negative control ($n = 5$ biologically independent samples examined over 3 independent experiments, statistical analysis by two-sided student's t-test, bars represent +/− SEM). **b** Analysis of C-circle abundance as in (**a**) in a panel of ALT-reliant cell lines in the presence or absence of control siRNA or siRNA targeting EXD2 as indicated. VA-13+hTel cells act as a telomerase-revertant control ($n = 2$ independent experiments, statistical ana-lysis by two-sided student's t-test, bars represent +/− SDM). **c** Western blotting depicting EXD2 expression in a panel of ALT-reliant cell lines. HeLa and VA-13+hTel cell lines act as a non-ALT and Telomerase-revertant control, respectively. α-Tubulin acts as a loading control. Verification of expression was carried out three times independently. **d** Quantification of the incidence of ALT-associated PML bodies in *EXD2*−/− U2OS compared to parental control cells as assayed by immunofluorescence staining using antibodies against TRF2 (red) and PML (green), DAPI (blue) acts as a nuclear stain ($n = 316$, 330 and 307 cells examined over 3 independent experiments, statistical significance confirmed by two-sided Mann−Whitney analysis, bars represent +/− SEM, scale bar = 10 µm). **e** Quantification of telomeric sister chromatid exchanges (T-SCEs) per chromo-some per cell in U2OS WT and *EXD2*−/− cells ($n = 75$ metaphase spreads from 3 independent experiments, statistical significance confirmed by two-sided Mann−Whitney analysis). **f** Quantitative FISH staining using strand-specific SpO and FITC-labelled PNA FISH probes specific to telomeric repeat sequences showing average telomere length ($n = 4187$ measurements per condition examined over 2 independent experiments, statistical significance confirmed by two-sided Mann−Whitney test, error bars represent +/−SEM). **g** Quantification of telomere length in U2OS WT and *EXD2*−/− cells by TeloSizer® analysis employing DNA combing and immuno-FISH staining to measure discreet telomere lengths ($n = 433$, 493 and 577 measurements examined over 3 independent experiments, statistical significance confirmed by two-sided Mann−Whitney analysis, bars represent +/− SEM). Source data are provided as a source data file.

(which we consider most likely) or a non-sister telomere (given that telomere clustering at APBs during ALT occurs throughout the cell cycle[43]). Moreover, as MUS81 can also aid the resolution of recombi-nation intermediates as part of the SMX (SLX1-4, MUS81-EME1, XPF-ERCC1) complex, SMX-dependent processing of migrating D-loops could also lead to T-SCE formation[39,44]. Accordingly, MUS81 depletion rescues both the excess conservative DNA synthesis and increased T-SCEs observed in EXD2-deficient cells, as well as limiting replication fork collapse and DSBs formation in the absence of EXD2. We therefore propose that MUS81 has a dual role upon EXD2-loss, promoting engagement of RAD52-independent BIR by cleavage of regressed forks at the telomere thus limiting RAD52-dependent BIR synthesis, while also likely promoting consecutive rounds of RAD52-independent BIR and/or generating T-SCEs as part of the SMX complex[42]. Importantly, T-SCEs are inversely correlated with ALT telomere synthesis[32] sug-gesting that loss of EXD2 may drive excessive engagement of BIR fol-lowed by MUS81-dependent D-loop cleavage, possibly driving C-circle formation. In further support of this notion, SMARCAL1-depletion also rescues the excess conservative DNA synthesis and elevated T-SCEs observed in EXD2-deficient cells indicating that regressed forks are the likely initial substrate that is processed for use by a RAD52-independent BIR mechanism, leading to either conservative DNA synthesis or T-SCEs.

Interestingly, EXD2-deficient U2OS cells display elevated fre-quencies of ALT telomere synthesis (as assayed by EdU incorpora-tion at telomeres and two-round segregated CO-FISH) yet have shorter average telomere length than WT cells. This may be due to engagement of RAD52-independent BIR which relies on MUS81 and does not require RAD52 for strand annealing to engage BIR. This RAD52-independent BIR may be, on average, less efficient than RAD52-dependent BIR, leading to overall shorter telomere length in cells primarily engaging this pathway. This could be due to the differences in factors required for its initiation and/or the template preferentially used by this pathway (i.e., intra- vs inter-chromosomal BIR). Alternatively, RAD52-independent BIR may result in the overall shorter telomere length in *EXD2*−/− cells due to attrition of telomeric sequences, i.e. excessive degradation of telomeric DNA at perturbed forks (we note that EXD2 protects nascent DNA at stalled forks in a pathway which is distinct from BRCA1/2-mediated fork protection[23]).

Recent studies have implicated FANCM in the ALT mechanism, with its loss resulting in hyperactivation of ALT-associated phenotypes coupled to telomere dysfunction[45–47]. It is unclear at present, however, whether EXD2 and FANCM operate in the same pathway, given that loss of BLM rescues viability of FANCM-depleted ALT + cells[46], with a separate study showing a synthetic sick relationship with FANCM[47]. Interestingly, SLX4IP, an SLX4 interacting factor was shown to counter BLM-mediated telomere clustering and dissolution of recombination intermediates in the ALT mechanism[36]. SLX4IP-deficient ALT + cells also display hyper-ALT activation and decreased telomere length, both reminiscent of EXD2-loss[36]. Crucially, SLX4-IP loss confers synthetic lethality in ALT-reliant cells upon co-depletion of SLX4, but not BLM[36]. In contrast, our data shows that EXD2-deficient U2OS cells display the opposite synthetic lethality phenotype, indicating that EXD2 loss confers hyperactivation of ALT and telomere length attrition via a parallel but independent pathway to SLX4IP.

Recently RAD51AP1, a factor which promotes homologous recombination at sites of active transcription and drives R-loop formation[48] was shown to promote the ALT mechanism[49] and may act as a strand annealing factor in RAD52-independent BIR[50,51], how-ever, its relationship with the role of EXD2 in the modulation of pathway choice at ALT Telomeres has not yet been elucidated.

In conclusion, our work identifies the early molecular events that orchestrate the use of ALT-associated homology-directed repair mechanisms at telomeres. We show that this process relies on the initial processing of stalled terminal replication forks and is directed by the action of the EXD2 nuclease as well as SMARCAL1 and MUS81 (Fig. 6d). Consequently, impaired processing of term-inal fork in ALT+ cells is associated with activation of conservative DNA synthesis.

## Methods

### Cell culture

HeLa and U2OS cells obtained from Dr. F. Esashi (University of Oxford, UK). U2OS stably expressing GFP were obtained from Prof. S. Jackson (University of Cambridge). The SV-40 large T-antigen transformed ALT cell lines GM-847, VA-13, and IMR-90 were dona-ted by Dr. A. Londoño-Vallejo (Institute Curie, Paris, France). The VA-13-h-Tel cell line that stably expresses human telomerase RNA component (hTERC) and human telomerase reverse transcriptase (hTERT) with reconstituted telomerase activity[27] was a gift from Prof J. W. Shay (UT Southwestern Medical Center, Dallas, TX, USA). The ALT+ liposarcoma Lisa-2 cells were kindly provided by Dr. D. Broccoli (Fox Chase Cancer Center, Philadelphia, PA, USA). The U2OS EJ2-GFP cells were a kind gift of Prof J. Stark (City of Hope, Department of Cancer Genetics and Epigenetics, Duarte, CA, USA). EXD2−/− U2OS cells were generated as described previously[23]. These were cultured in Dulbecco's modified Eagle's medium (DMEM) supplemented with 10% foetal bovine serum (FBS) and standard antibiotics. U2OS cells overexpressing cyclin E in a tetracycline-dependent manner and *RAD52*−/− cells generated by CRISPR-Cas9 in this background, were a kind gift from Prof T. Halazonetis (University of Geneva, Switzerland). These were cul-tured in DMEM supplemented FBS, standard antibiotics, puromycin

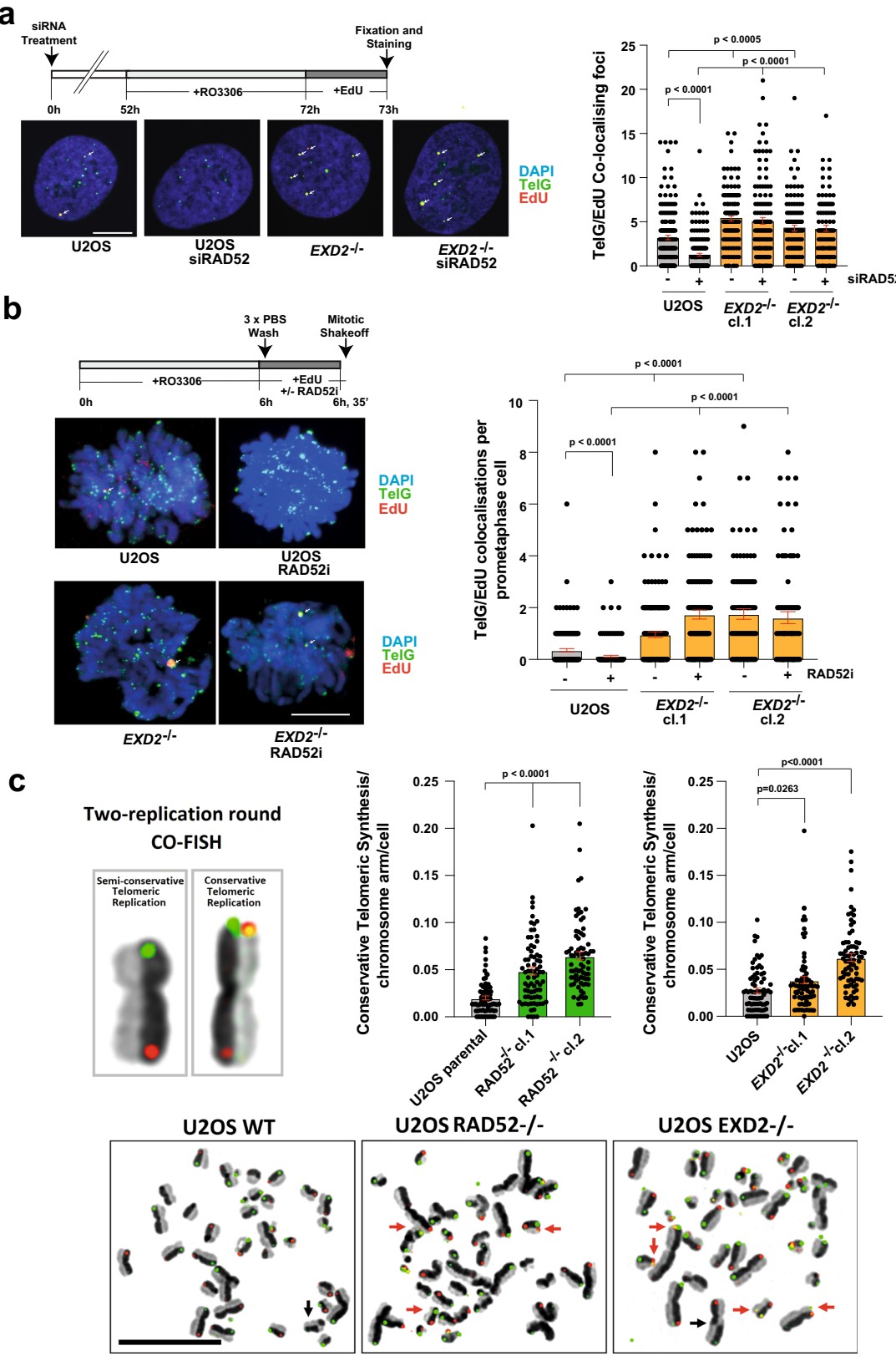

(1 μg/ml) and tetracycline (2 μg/ml) as previously described[31]. U2OS cells stably expressing GFP-EXD2, FLAG-HA WT or nuclease-dead EXD2 (generated previously[22,23]) were cultured in DMEM with 10% FBS and antibiotics, supplemented with either 500 μg/ml G-418 for GFP-expressing cells (Thermofisher) or 0.5 μg/ml Puromycin (GIBCO) for FLAG-expressors. Pools of *EXD2*−/− U2OS cells stably

expressing FLAG-HA EXD2 WT or nuclease dead protein were generated by transfection with plasmid constructs encoding WT or nuclease dead EXD2 in the pHAGE-N-Flag–HA vector backbone and selection with puromycin (1 μg/ml) followed by culture in 0.5 μg/ml Puromycin in DMEM supplemented with 10% FBS and standard antibiotics.

**Fig. 4 | EXD2-deficiency switches pathway choice from canonical HR towards BIR at telomeres in ALT-positive cells. a** Immuno-FISH in WT and *EXD2*−/− U2OS cells synchronised to G2 by RO-3306 treatment and labelled with EdU as indicated. EdU incorporation was determined by Click-iT staining (red) and telomeres marked by a telomere-specific PNA FISH probe (TelG, green) with co-localisation indicated by arrows. Cells were treated with control siRNA or siRNA targeting RAD52 as indicated. DAPI (blue) acts as a DNA stain ($n$ = 164, 155, 120, 130, 108 and 99 cells examined over 3 independent experiments, statistical significance was determined by two-sided Mann–Whitney test, bars represent +/− SEM, scale bar = 10 μm). **b** Immuno-FISH analysis in WT or *EXD2*−/− U2OS cells synchronised to G2 by RO-3306 followed by release to synchronised prometaphase in the presence of EdU as indicated. EdU incorporation was determined by Click-iT staining (red) and telomeres marked by a telomere-specific PNA FISH probe (TelG, green) with co-localisations indicated by arrows. DAPI (blue) acts as a DNA stain. Cells were treated with DMSO, or RAD52-inhibitor ($n$ = 140, 157, 144, 118, 86 and 83 prometaphases examined over 3 independent experiments, statistical significance was determined by two-sided Mann–Whitney analysis, bars represent +/− SEM, scale bar = 10 μm). **c** Quantification of conservative telomeric synthesis in WT vs *EXD2*−/− U2OS cells and *RAD52*−/− Cyclin E-overexpressing U2OS vs parental control cells as determined by two-replication round segregated CO-FISH using SpO and FITC-labelled PNA FISH probes (red and green signals, respectively) specific to the telomeric repeat sequences ($n$ = 75 metaphases examined over 3 independent experiments, statistical analysis was carried out by two-sided Mann–Whitney analysis, bars represent +/− SEM, scale bar = 10 μm). Source data are provided as a source data file.

## Cell synchronisation, EdU labelling and isolation of prometaphase cells and metaphase chromosomes

For synchronisation, 7 μM RO-3306 (Sigma) was added to cells for the indicated times to allow accumulation of cells in the late G2 phase of the cell cycle. All EdU incorporations described were carried out using 20 μM EdU in the presence or absence of Mirin (50 μM) or Rad52-inhibitor (AICAR, 40 μM), as indicated. To arrest cells in metaphase, 0.1 μg/ml Colcemid (GIBCO BRL-Karyomax) was added to cells for 1 h following release from RO-3306 treatment as indicated. Synchronised prometaphase cells were harvested by mitotic shake-off and spun onto poly-l-lysine slides at 237 × *g* for 2 min.

For analysis of EdU incorporation in metaphase chromosomes, cells were collected by mitotic shake-off and swollen in KCL for 15 min at 37 °C before dropping on to poly-l-lysine slides and centrifugation at 237 × *g* for 2 min. Isolated prometaphase cells or metaphase chromosomes were then pre-extracted for 1 min in 0.5% Triton-100X at room temperature followed by PBS rinses and fixation in 4% PFA for 15 min. EdU incorporation was performed using the Click-iT EdU Alexa Fluor 555 Imaging kit (Thermofisher) following the manufacturers recommendations. After EdU labelling, coverslips were washed 3× 5 min in PBS before fixation in 4% PFA for 20 min. After PBS rinses, coverslips were then dehydrated with 70%, 85%, 100% EtOH series followed by FISH hybridisation as described below.

## Colony formation assays

Colony formation assays were performed by seeding 500 cells per well in 6-well plates 72-h post-treatment with the first pulse of either control siRNA or siRNA targeting various genes (as described). Colonies were stained by methylene blue staining 10 days later and counted using an Oxford Optronix Gelcount machine and Gelcount software. Colony numbers in each cell line were measured relative to normalised controls (siLuciferase treated controls in each case) and the percentage surviving colonies vs control were calculated.

## siRNA treatment

siRNAs employed were as follows:

BRCA2 (ON-TARGETplus SMART pool, L-003462-00-0005, Dharmacon), BLM (5′-GCUAGGAGUCUGCGUGCGA-3′, or ON-TARGETplus SMARTpool (L-007287-00-0005), Dharmacon) DNA2 (ON-TARGETplus SMARTpool (L-026431-01-0005), Dharmacon or s531517 Silencer Select siRNA, Thermo-Fisher), EXD2 (5′-CAGAGGACCAGGUAAUUUA-3′), MRE11 (5′-GGAGGUACGUCGUUUCAGA-3′), MUS81 (ON-TARGETplus SMARTpool (L-016143-01-005), Dharmacon or s37039 Silencer Select siRNA, Thermo-Fisher), POLD3 (s21045 silencer select siRNA (Thermo-Fisher) or ON-TARGETplus SMARTpool (L-026692-01-0005), RAD51 (5′-GAGCUUGACAAACUACUUCUU-3′), RAD52 (ON-TARGETplus Human SMARTpool (L-011760-00-0005) Dharmacon, SLX4-s39053 Silencer Select siRNA, Thermo-Fisher), SMARCAL1 (ON-TARGETplus SMARTpool (L-013058-00-0005), Dharmacon or s531776 Silencer Select siRNA, Thermo-Fisher). siRNA targeting luciferase − 5′-CGUACGCGGAATACTTCGA-3′ was used as control siRNA.

Oligonucleotides were transfected using HiPerfect reagent (QIAGEN), in line with the manufacturer's protocol.

## Cell lysis and immunoblotting

Cells were lysed using (9 M urea, 50 mM Tris–HCL, pH 7.5, 150 mM β-mercaptoethanol) followed by sonication using a Soniprep 150 sonicator. Samples were resolved by SDS-PAGE and transferred to PVDF or nitrocellulose. Protein concentrations were determined by Bradford assays via spectrophotometry using a DeNovix DS-11 FX + spectrophotometer. Immunoblots were carried out using the indicated antibodies: α-Tubulin (Sigma, B-5-1-2; T5168, 1:100,000), BLM (Bethyl, A300-110A 1:2000), DNA2 (Abcam ab962488, 1:1000), EXD2 (Sigma, HPA005848, 1:1000), MCM2 (Abcam, ab4461, 1:10,000), MRE11 (Abcam, ab214, 1:1000), MUS81 (Abcam, ab14387 1:1000), POLD3 (Abnova, H00010714-M01, 1:500), RAD52 (28045-1-AP, Proteintech, 1:2000), SMARCAL1 (Santa Cruz, sc-376377 1:1000), SLX4 (University of Dundee, DU16029, 1:200), Vinculin (Thermo-Fisher, MA5-11690, 1:1000).

## Alt-EJ GFP reporter assay

U2OS EJ2-GFP cells[52] were treated with two pulses of control siRNA or siRNA targeting EXD2, MRE11 or BRCA1 (as indicated) and transfected using Amaxa nucleofection with an I-SceI expression vector (pCMV-I-SceI) or a vector expressing mCherry fluorescent protein (pmCherry-C1) 48 h post-treatment with the first pulse of siRNA. 72 hours after I-SceI transfection, cells were trypsinised and analysed by flow cytometry (BD LSR II, $2 \times 10^4$ cells per experimental condition). GFP-positive cells per 1000 mCherry-positive cells was determined using BD FACS DIVA software, with data related in each experiment to the siControl treated. Statistical significance was determined by student's t test.

## Immunofluorescence and immuno-FISH staining

For immunofluorescence staining, asynchronous cells grown on coverslips were fixed with 4% PFA for 10 min at room temperature rinsed twice in PBS and permeabilised with 0.2% Triton X-100 in PBS for 10 min at room temperature. In some instances, cells were fixed with ice cold Methanol min or 250 mM HEPES, 1x PBS, pH 7.4, 0.1% Triton X-100, 4% PFA for 20 min on ice or were pre-permeabilized on ice using either 0.2% Triton X-100 in PBS or 10 mM PIPES. pH 7.0, 100 mM NaCl, 300 mM sucrose, 3 mM MgCl₂, 0.7% Triton X-100) for 10 min prior to fixation with 4% PFA as above. Coverslips were then rinsed in PBS and blocked with 10% FBS in PBS for 30 min before incubation with primary antibodies in 0.1% FBS in PBS for 1–12 h at room temperature, then washed 4× for 5 min in PBS with subsequent incubation with secondary antibodies (Alexa-Fluor 488, 555, 568 or 647 (Molecular Probes or Invitrogen)). Slides were then washed 4× for 5 min in PBS and either mounted with Vectashield with DAPI (Vector, H-1200-10) or fixed in 4% PFA (Thermofisher) in PBS for 20 min at room temperature to be processed for immuno-FISH staining.

For immuno-FISH, fixed coverslips prepared as above were dehydrated using 70%, 85% and 100% ethanol series before PNA-

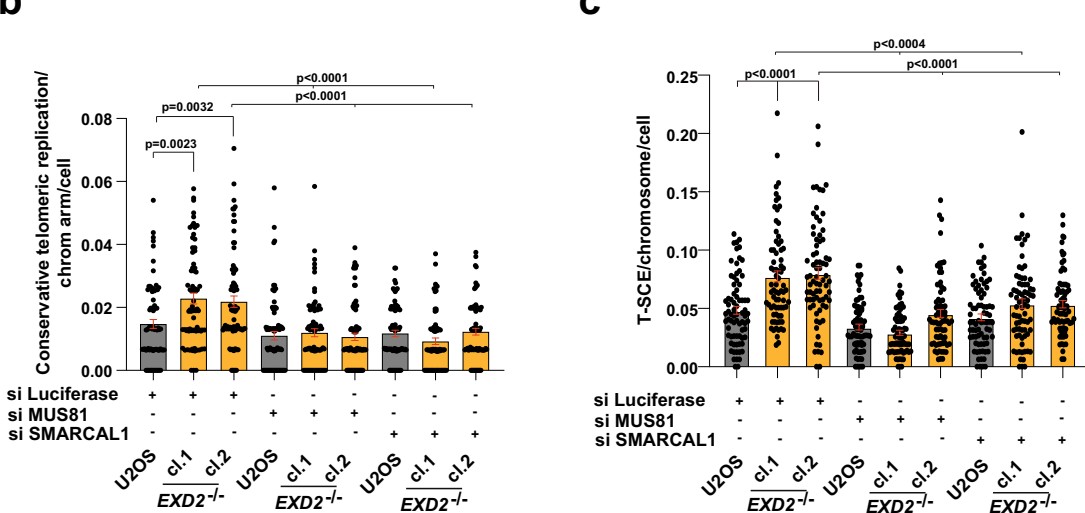

FISH probe hybridisation using a FITC-labelled PNA probe specific to the G-rich telomeric sequence at 5 nM (PN-TG011-005, PNABio) diluted in 20 mM $Na_2HPO_4$, 20 mM Tris–HCl, pH 7.5, 60% formamide, 2XSSC and 0.1 µg/ml salmon sperm DNA at 80 °C for 6 min before cooling to room temperature for 2 h. Slides were then washed 2×10 min in 2XSSC, 0.1% Tween-20 at 60 °C followed by

2×15 min washes in PBS at room temperature before mounting in Vectashield with DAPI.

Primary antibodies employed for immunofluorescence were as follows: γH2AX (JBW301, Millipore, 1:500), 53BP1 (MAB3802, Millipore, 1:1000), MRE11 (ab214, Abcam, 1:500), MUS81 (sc53382, Santa Cruz Biotechnology, 1:200), PML (E-11 sc-377390, Santa Cruz Biotechnology,

**Fig. 5 | EXD2-deficient cells engage BLM-dependent conservative mitotic telomere synthesis and form T-SCEs in a SMARCAL1- and MUS81-dependent manner. a** Immuno-FISH staining in WT and *EXD2*[−/−] U2OS synchronised prometaphase cells treated with siRNA targeting BLM, SLX4, SMARCAL1 or control siRNA. EdU incorporation was determined by Click-iT staining (red) and telomeres marked by a telomere-specific PNA FISH probe (TelG, green), DAPI (blue) acts as a nuclear stain (*n* = 80, 80, 76, 75, 55, 55, 51, 42, 54, 56, 62 and 48 prometaphases examined over 2 independent experiments, statistical significance was determined by two-sided Mann–Whitney analysis, bars represent +/− SEM, scale bar = 10 μm).
**b** Quantification of conservative replication in U2OS WT and *EXD2*[−/−] as determined by two-replication round segregated CO-FISH using SpO and FITC-labelled PNA

FISH probes specific to the telomeric repeat sequences. Cells treated with siRNA targeting SMARCAL1 or MUS81, or were treated with control siRNA targeting Luciferase, as indicated (*n* = 75 metaphases examined over 3 independent experiments, statistical significance was determined by two-sided Mann–Whitney analysis, bars represent +/− SEM). **c** Quantification of T-SCEs in U2OS WT and *EXD2*[−/−] as determined by one-replication round CO-FISH using SpO and FITC-labelled PNA FISH probes specific to the telomeric repeat sequences. Cells were treated with siRNA targeting SMARCAL1 or MUS81, or were treated with control siRNA targeting Luciferase, as indicated (*n* = 75 metaphases examined over 3 independent experiments, statistical significance was determined by two-sided Mann–Whitney analysis, bars represent +/− SEM). Source data are provided as a source data file.

1:500 or Ab96051, Abcam, 1:500), POLD3 (Abnova, H00010714-M01, 1:500), RAD51 (Bioacademia, 70-002; 1:500), RAD52 (Sheep, a kind gift from Prof T. Halazonetis University of Geneva, Switzerland, 1:100), SLX4 (University of Dundee, DU16029, 1:200), SMARCAL1 (Santa Cruz, sc-376377 1:500) TRF1 (Rabbit, #6839 a gift from Prof J. Karlseder-The Salk Institute for Biological Studies. La Hoya, USA) and TRF2 (A300-796A-T, Thermo-Fisher, 1:500, PA1-41023. Thermo-Fisher 1:500) or Rabbit #6841 1:500, also kindly donated from Prof J. Karlseder), RPA1 (Ab-3; Calbiochem 1:1000). Images were acquired using an Advanced Spinning Disc confocal microscope using 3i acquisition software and advanced spinning disc microscope or a Zeiss Axio-Imager Z1 (Zeiss) microscope equipped with a CCD camera. Image analysis was carried out in FIJI or Metasystems Isis Software.

## PLA assays
Proximity ligation assays were carried out in WT U2OS cells or cells stably overexpressing GFP-EXD2 or FLAG-HA-EXD2. For PLA experiments described, cells grown on coverslips were fixed with 4% PFA in PBS for 10 min at room temperature followed by 3× rinses with PBS and permeabilisation with 0.2% Triton-100X in PBS for 10 min at room temperature followed by blocking with 10% FBS in PBS for 30 min at room temperature. For PLA reactions using antibody pairs for GFP and TRF2, as well as FLAG and TRF2, cells were instead fixed in 100% MeOH at −20 °C for 20 min before proceeding directly to 3× PBS rinses and blocking.

After blocking, primary antibody pairs were diluted in 0.1% FBS in PBS for 1 h at room temperature. Coverslips were then washed 2× in PBS before performing proximity ligation reaction using the Duolink In Situ Red Starter kit (Sigma Aldrich) following the manufacturer's instructions. Coverslips were mounted using Vectashield with DAPI and microscopy carried out using a 3i Advanced spinning disc microscope or a Zeiss LSM 510 microscope and ×63 objective. Image analysis was carried out using FIJI.

Antibodies used for PLA were as follows:

53BP1 (MAB3802, Millipore, 1:1000), FLAG (Sigma M2, 1:500), GFP (Roche 11 814 460 001, 1:500), TRF1 (Rabbit, #6839 from Prof J. Karlseder 1:200), TRF2 (Rabbit #6841 from Prof J. Karlseder, 1:500), TRF2 (A300-796A-T 1:200)

## DNA fibre analysis
Exponentially growing cells were labelled with 25 mM IdU and 125 mM CldU (as indicated, followed by cell lysis, fibre spreading with staining carried out as previously described[23]. Images were obtained using a Leica SP8 confocal microscope using a ×63 oil objective. Image analysis was carried out using FIJI (ImageJ) Software.

## C-circle assay
DNA extraction for C-circle assays was carried as previously described[53]. Cells were lysed at 37 °C in 2% SDS, 50 mM Tris-pH 7.5, 20 mM EDTA and 200 μg/ml Pronase Protease (Sigma) followed by DNA concentration by Sodium-Acetate/Ethanol precipitation. DNA quantification was carried out using the Qbit dsDNA BR kit (thermofisher) and DeNovix DS-11 FX + spectrophotometer following the

manufacturers recommendations. For C-circle reactions, 32 ng of extracted DNA was added to PCR master mix containing 0.2 mg/ml BSA, 0.1% Tween-20, 4 mM DTT, 1 mM each in the presence or absence of Phi-29 polymerase (NEB, 3.75U/32 ng DNA) and 1x Phi-29 reaction buffer (NEB) with rolling circle amplification carried out as previously described[53]. To quantify C-circle abundance, each reaction was diluted with 100 μl 2X SSC and slot blotted using Bio-Rad Slot blot apparatus onto Hybond-N + membranes (Amersham) which were UV crosslinked with autocrosslink settings (120 mJ/cm²) of Stratalinker 1800 (Stratagene) apparatus, hybridised overnight at 37 °C with $^{32}P(CCCTAA)_3$ labelled telomere probe in PerfectHyb buffer (Sigma) before washing 4× at 37 °C in 0.5X SSC, 0.1% SDS buffer and imaging using Typhoon FLA 9000 (GE Healthcare). Densitometric analysis was performed using FIJI software.

## T-SCE and segregated two-replication round CO-FISH staining
Cells in culture were labelled with 7.5 μM BrdU and 2.5 μM BrdC for either one or two cell cycles (18–20 or 36–40 h, respectively) followed by incubation for 1 h with 0.1 μg/ml Colcemid (GIBCO BRL-Karyomax). Cells were trypsinised and harvested by centrifugation (10 min, 200 × *g*). Cells were swelled with 75 mM KCL at 37 °C for 20 min and fixed with 3:1 Methanol:Acetic acid (ice cold). Chromosome preparations were dropped onto wet glass slides that were air dried and aged overnight.

CO-FISH staining was performed as described previously[16]. Briefly, slides were stained with Hoechst 33258 (0.5 μg/ml, Sigma) and incubated in 2× SSC for 15 min at RT, then treated with 0.5 mg/ml RNase A (in 1x PBS or 2x SSC) for 1 h at 37 °C. Consequently, slides were exposed to 365-nm UV light (Stratalinker 1800 UV irradiator) for 45 min. BrdU/C labelled DNA was digested with exonuclease III diluted in 5 mM DTT, 5 mM MgCl₂, and 50 mM Tris−HCl, pH 8.0 for 15–30 min at 37 °C. Sides were then dehydrated by cold ethanol series (70, 85 and 100%) and air-dried.

PNA FISH staining was carried out using probes specific for C- and G-rich telomere repeats (FITC-(CCCTAA)₃ and Cy3-(TTAGGG)₃ (Bio-Synthesis or Panagen-South Korea) diluted in 10 mM Tris−HCl, pH 7.2, 70% formamide, 0.5% blocking reagent (Roche). The first probe (0.8 μM, Cy-3) was hybridised for 1 h at RT in humidity, rinsed in wash I solution (70% formamide, 10 mM Tris−HCl, pH 7.2, and 0.1% BSA) for 5 min followed by hybridisation of the second probe (0.5 μM, FITC). Slides were washed (2 × 15 min) in Wash I and 3 ×5 min with Wash II (0.1 M Tris−HCl, pH 7.4, 0.15 M NaCl, 0.08% Tween-20) before dehydration with cold ethanol series and mounting in Vectashield with DAPI. Images were acquired with an Axio Imager Z1 Zeiss microscope with a ×63 objective and analysed using MetaSystems Isis software. Two replication-round Segregated CO-FISH is presented in detail in Supplementary Fig. 5.

## Quantitative PNA-FISH
Quantitative PNA-FISH staining was carried out as previously described[16], using either G- or C-rich probes recognising the telomeric repeats (as above). Telomere fluorescence intensity was estimated using the MetaSystems Isis/Telomere software which normalised

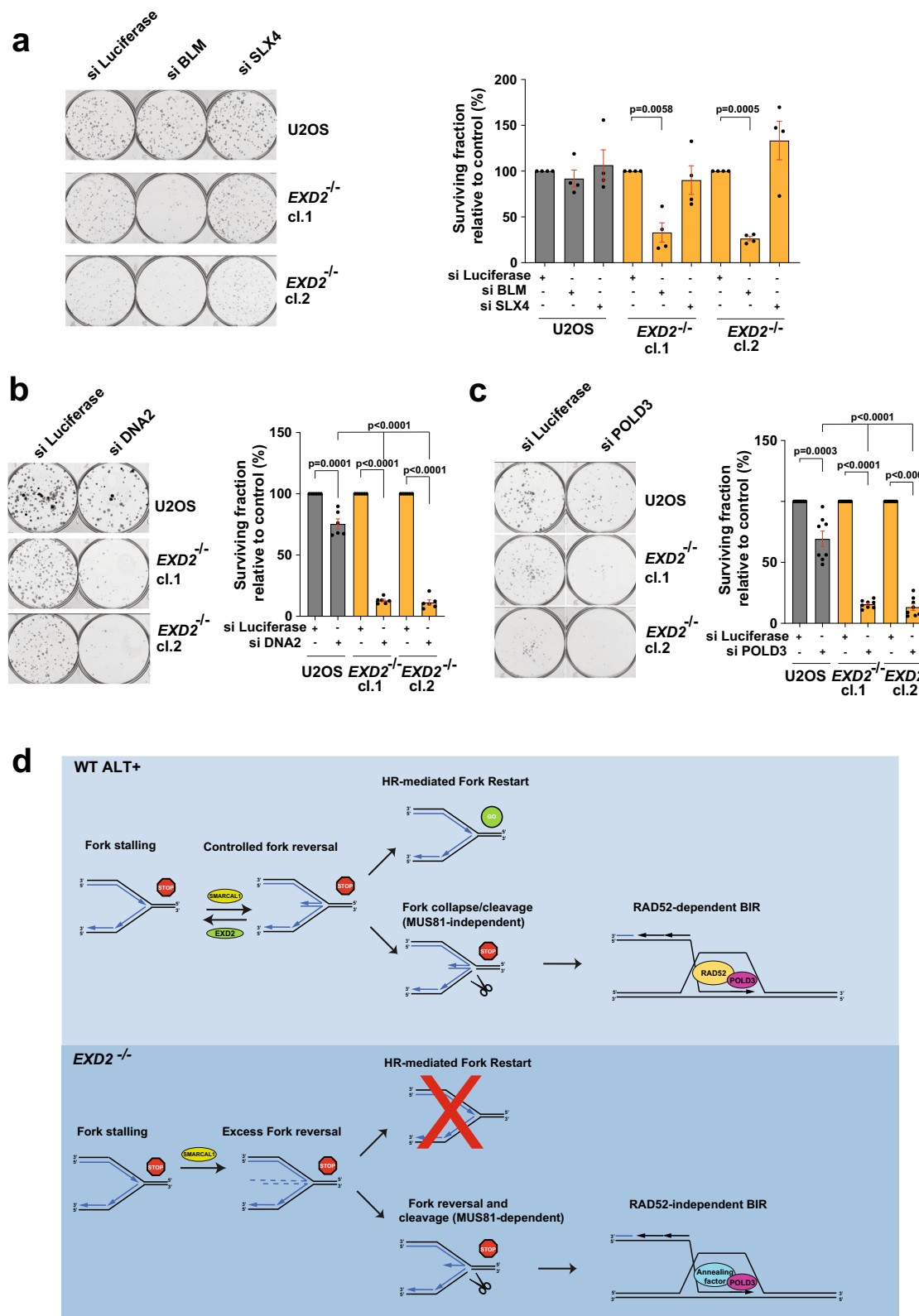

signals from each metaphase spread to background staining, exposure time normalization was also included. Analysis of unreplicated sister telomeres was carried out by staining with PNA-FISH probes specific to the G- and C-rich telomeric repeats (as describe above) with analysis of chromosome arms displaying extreme fluorescence intensity discrepancies between sister chromatids at both fluorescent channels.

**TeloSizer® analysis**

Measurement of telomere lengths in U2OS WT and EXD2-deficient cells was carried out in collaboration with Genomic Vision. Briefly, cells were trypsinised, washed 1× in PBS and 50,000 cells embedded into low melting point agarose plugs using the FiberPrep DNA extraction kit and processed by the Genomic Vision TeloSizer® pipeline. Briefly,

**Fig. 6 | EXD2-deficient U2OS cells are synthetic sick upon depletion of factors that promote BIR-mediated ALT DNA synthesis. a** Colony formation assays in WT U2OS and *EXD2*⁻/⁻ cells treated with siRNA targeting BLM, SLX4 or control siRNA (Luciferase). Surviving fraction relative to control is quantified (*n* = 4 biologically independent samples examined over 2 independent experiments, statistical significance was determined by two-sided student's t-test, bars represent +/−SEM). **b** Colony formation assays in WT U2OS and *EXD2*⁻/⁻ cells treated with siRNA targeting DNA2 or control siRNA. Surviving fraction relative to control is quantified (*n* = 6 biologically independent samples examined over 3 independent experiments, statistical significance was determined by two-sided student's t-test, bars represent +/−SEM). **c** Colony formation assay in WT U2OS and *EXD2*⁻/⁻ cells treated with siRNA targeting POLD3 or control siRNA, as indicated. Surviving fraction relative to control is quantified (*n* = 8 biologically independent samples examined over 2 independent experiments, statistical significance was determined by two-

sided student's t-test, bars represent +/−SEM). Source data are provided as a source data file. **d** Model: WT U2OS cells restart stalled replication forks at the telomere by HR-mediated fork restart or employ RAD52-dependent break induced replication upon replication fork collapse/cleavage (MUS81-independent). RAD52 acts as a strand annealing factor to promote POLD3-dependent conservative DNA synthesis. In the absence of EXD2, excessive replication fork regression mediated by SMAR-CAL1 leads to fork degradation and fork collapse by MUS81-dependent nucleolytic cleavage. This results in a fork conformation that is preferentially processed by the RAD52-independent arm of the ALT BIR mechanism, whereby an as-yet unidentified annealing factor promotes POLD3-dependent conservative DNA synthesis resulting in poor telomere elongation and the presence of T-SCEs. In the absence of factors that promote BIR-mediated ALT synthesis, collapsed replication forks at the telomere in EXD2-deficient cells cannot be efficiently repaired.

Genomic Vision carried out genomic DNA extraction for DNA combing using the FiberPrep DNA extraction kit (Genomic Vision) according to the manufacturer's instructions. Combing was then carried out on silanised glass slides (Genomic Vision) using a FiberComb molecular combing machine (Genomic Vision). Slides were then stained for total genomic DNA and hybridised with a FISH probe recognising telomeric DNA. Telomere lengths at the ends of chromosomes were then measured using a FiberVision automated scanner (Genomic Vision).

### Statistical analysis
Statistical significance was determined using Graphpad PRISM 9 software by two-tailed Mann–Whitney analysis or unpaired two-tailed t-test analysis.

### Reporting summary
Further information on research design is available in the Nature Portfolio Reporting Summary linked to this article.

## Data availability
Raw data for imaging experiments associated with this manuscript are available as Mendeley dataset with https://doi.org/10.17632/dmk886xwxw.1 [https://data.mendeley.com/datasets/dmk886xwxw]. Source data are provided with this paper.

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

## Acknowledgements

We thank Dr. F. Esashi, Dr. A. Londoño-Vallejo, Prof. J. W. Shay, Dr. D. Broccoli, Prof. J. Stark, Prof. S. Jackson for cell lines, Prof. T. Halazonetis for cell lines and antibodies specific for RAD51 and RAD52, and Prof. J. Karlseder for antibodies recognizing TRF1 and TRF2. We thank Genomic Vision for carrying out the TeloSizer® analysis as part of their Triathlon programme. We thank Dr. A. Kanellou, Dr. C. Smith and Dr. P. Martin for technical assistance. We also thank Dr. M. Douglas, and the members of Niedzwiedz's team for their comments on the manuscript and the S.G. team for input on experimental design. Work in W.N.'s lab was funded by ICR intramural grant and Cancer Research UK programme (A24881); R.B. was supported by the ICR Dean Award. Work in S.G.'s lab was supported by BRFAA Intramural Funds.

## Author contributions

R.B. carried out IF assays, PLA, Immuno-FISH and C-circle assays, siRNA knockdown experiments, WBs, colony survival assays and EdU/BIR analyses in metaphase, prometaphase and G2 cells; V.C. performed IF, FISH, Immuno-FISH, C-circle assays and the validation of segregated CO-FISH assays; J.N. carried out PLA, DNA fibre, C-circle assay and Alt-EJ reporter assays; E.D. carried out IF, FISH, Q-FISH analyses, T-SCE, and segregated CO-FISH assays; T.E. executed IF, FISH, Immuno-FISH, and T-SCE assays; M.K. carried out segregated CO-FISH analyses. R.B., S.G. and W.N., conceived the project, contributed to experimental design, and wrote the manuscript.

## Competing interests

W.N. is a named inventor on a patent describing the use of EXD2 inhibitors and stands to gain from their development as part of the ICR "Rewards to Inventors" scheme; and was a consultant for MNM Bioscience. The remaining authors declare no competing interests.
