## [Peer Review File · Nature Communications]

Pathway choice in the alternative telomere lengthening in neoplasia is dictated by replication fork processing mediated by EXD2's nuclease activityREVIEWER COMMENTS

Reviewer #1 (Remarks to the Author):

Comments to Broderick et al:

In the submitted manuscript (ms), Broderick et al reported that EXD2 localizes to telomeres in the ALT cells. Deficient EXD2 cells (using either siRNA targeting EXD2 and/or two genetic EXD2 knockout clones) manifested many dysfunctions/defects of ALT telomeres. Most important, they found that EXD2 might affect the sub-pathway selection in the ALT pathway. Unfortunately, many of the data presented in this ms are of poor quality. Therefore, it is very difficult to evaluate the potential important findings claimed by the authors. Therefore, I recommend rejection in its current form.

Specific points on the data quality in different figures: In general, many of the Western Blots (WB) are of poor quality. Some of the loading controls fluctuate too much (e.g., S1b, S1f, S2d etc). Therefore, it is very difficult to judge the changes or lack of changes of protein of interest. The major findings also need to be repeated in another ALT cell line.

Fig 1: The major weakness of the EXD2 telomere localization is that it lacks data using the antibody (Ab) recognizing the endogenous EXD2. If the EXD2 Ab is not suited for the immunofluorescence assay, the authors need to try other assays to demonstrate its recruitment to ALT telomeres. Since the authors rely heavily on the tagged EXD2 (GFP-EXD2 and FLAG-EXD2), they need to first demonstrate the expression level of these tagged proteins relative to the endogenous EXD2. The authors did attempt to address this for the FLAG-EXD2 in Fig S1b. However, the level for the endogenous EXD2 is too weak to be used to estimate the relative level of the exogenous FLAG-EXD2. The authors also need to quantify them. If the level is more than 10-fold, they should use viral vector to bring down the expression level of the tagged proteins to the endogenous EXD2. They also need to do the same for the GFP-EXD2. Secondly, they need to use the correct controls. For example, the appropriate control cell line for the GFP-EXD2 should be U2OS expressing GFP only rather than just the parental U2OS.

Fig 2: The difference between wild-type (WT) U2OS and the two EXD2 knockout clones (EXD2^{-/-}) are very subtle in some of the experiments (2b, 2c, 2d). Therefore, all the data using the EXD2^{-/-} need to be rescued using the WT-EXD2.

Fig 3: The difference between wild-type (WT) U2OS and the two EXD2 knockout clones (EXD2^{-/-}) are very subtle in some of the experiments (3a, 3b, 3d). Therefore, all the data using the EXD2^{-/-} need to be rescued using the WT-EXD2. In addition, the figures 3e and 3f and their legends are not matching. The author need perform the TRF assay to quantify the telomere length.

Fig 4: The difference between wild-type (WT) U2OS and the two EXD2 knockout clones (EXD2^{-/-}) are very subtle in some of the experiments (4a to 4c). Therefore, all the data using the EXD2^{-/-} need to be rescued using the WT-EXD2. The results between siRad52 and Rad52i are inconsistent for cl.1. Why is that?

Fig 5: The difference between wild-type (WT) U2OS and the two EXD2 knockout clones (EXD2^{-/-}) are very subtle in some of the experiments (4a to 4c). Therefore, all the data using the EXD2^{-/-} need to be rescued using the WT-EXD2. Are there cell cycle and/or cell death changes when co-depleted with siBLM, siSLX4, siMUS82 and siSMARCAL1?

Fig 6: Both EXD2^{-/-} clones seem to be quick sick on their own (Figs 6, S4, S5). Are there any cell cycle and growth rate changes by themselves? In addition, all the experiments with positive results using siRNA need to be repeated with the second independent siRNA.

Fig S1: The WB for SMARCAL1 is of poor quality.

Fig S2: The WB for Rad52 is of poor quality.

Fig S4: The WB for SLX4 is of poor quality. SLX4 seems to be drastically reduced in the EXD2^{-/-} cells. Why is that?

Fig S5: Human DNA2 has 1060 aa. The size in the WB (above 170 KDa) seems to be very high relative to its predicated size (5a). Furthermore, previous publications showed that both recombinant and endogenous DNA2 runs around 130 KDa on a SD-PAGE (PMID: 23604072 and 31216032). Please make sure that the DNA2 Ab used in the WB recognizes the right protein.

Minor points:

Line 43: Dagg et al (2017, PMID 28636942) showed that a small percentage of TLM negative tumors can survive >200 population doublings with ever-shorter telomeres. Therefore, the statement may not be completely correct.

Line 48-50: Inappropriate citations. Recommended citations: PMID – 34069193, 26654051

Line 50-51: Imprecise statement. For example, prior to chemotherapy, the overall survival rate for osteosarcoma was a dismal 20%. With chemotherapy, the overall 5-year survival rate increase to 60%–70% (PMID: 21559216).

Line 301-304: Pan et al (2017, PMID: 28673972) first reported the involvement of FANCM in the ALT pathway. In the same study, they reported synthetic lethal interaction between FANCM and BLM.

Reviewer #2 (Remarks to the Author):

Cancer cells must maintain their telomere length in order to support their unlimited proliferation. About 10-15% of human cancers rely on a homologous recombination-based mechanism, known as alternative lengthening of telomeres (ALT), to maintain their telomeres. ALT cells exhibit a number of hallmarks (APBs, T-SCEs, telomere heterogeneity and C-circles) and display a heightened level of telomere replication stress. ALT is thought to occur through a RAD52-dependent and a RAD52-independent BIR pathway. Many proteins that are involved in HR and replication stress response are associated with ALT telomeres and play a role in ALT.

The manuscript by Broderick et al investigated the role of EXD2, a nuclease that has been implicated in HR, fork restart and fork protection, in ALT cells. The authors showed that EXD2 is recruited to ALT telomeres and regulates ALT hallmarks. They went on to further characterize RAD52-independent telomere DNA synthesis in EXD2^{-/-} cells. They concluded that EXD2 dictates pathway choice in ALT cells.

There are several major issues associated with this manuscript.

Major issues,

1. The authors fell short of demonstrating that EXD2 dictates the pathway choice at ALT telomeres. Although they showed that telomere DNA synthesis in EXD2^{-/-} cells is RAD52-independent, EXD2^{-/-} cells exhibit telomere shortening. The latter is consistent with the notion that RAD52-dependent BIR is impaired at ALT telomeres in EXD2^{-/-} cells. It is possible that EXD2 does not regulate the pathway choice per se but rather is engaged in RAD52-dependent BIR and that EXD2^{-/-} cells simply rely on RAD52-independent BIR for telomere synthesis because of impaired RAD52-dependent BIR.
2. The authors show that EXD2 loss leads to reduced levels of SLX4 (Fig. S4a), MUS81 (Fig. S4b), and MRE11 (Fig. S2d). Given that SLX4, MUS81 and MRE11 are known to be engaged in ALT, their finding raises a concern as to whether or what portion of the observed phenotypes in EXD2^{-/-} cells are due to EXD2 itself.
3. Previously, Zhang et al (2019) reported that RAD51 and MRE11 inhibit RAD52-independent BIR-mediated C-circles. Does EXD2 loss affect the level of RAD51? Do RAD51 and MRE11 inhibit RAD52-independent BIR-mediated telomere DNA synthesis and C-circles in EXD2^{-/-} ALT cells?
4. The authors claimed that there is an epistatic relationship between EXD2, RAD51 and MRE11 in repair of T-DSBs in ALT and that RAD51- and MRE11-dependent repair pathways can be ruled out as the main repair pathways utilized in ALT cells in the absence of EXD2. This claim was based solely on the 53BP1/ALT telomere colocalization experiment and therefore it is not sufficiently supported. The authors should at least investigate if EXD2 loss affects association of HR proteins with ALT telomeres and vice versa.

5. The authors claimed that EXD2 nuclease activity is required for the pathway choice, but they fell short of testing EXD2 nuclease-dead mutants in RAD52-independent BIR in ALT.

6. The authors suggest that EXD2 loss leads to excess replication fork reversal (by SMARCAL1) at ALT telomeres, which are cleaved by MUS81 and repaired by RAD52-independent BIR. Does EXD2 loss lead to a reduction in RAD52 association with ALT telomeres? Does EXD2 loss lead to an increase in telomere association of SMARCAL1, MUS81 and POLD3?

7. In their model, the authors proposed that HR-mediated fork restart at ALT telomeres is disabled in the absence of EXD2 but they did not provide supporting data.

Minor issues:

1. Ref 16 on line 67 is incorrectly cited.
2. Line 171, "than" is misspelled.
3. Line 161, Fig. 2d is incorrectly cited.
4. Fig. S4a, the SLX4 blot is of poor quality. SLX4 does not appear to be knocked down in the last two lanes.

Reviewer #3 (Remarks to the Author):

The authors describe the function of EXD2 in the ALT mechanism and attempt to differentiate its relative contributions to RAD52-dependent and RAD52-independent telomere synthesis. The authors found that EXD2 colocalizes with ALT telomeres and APBs. Additionally, they thoroughly demonstrate that EXD2 KO cells have enhanced ALT phenotypes (TIF formation, RPA colocalization to telomeres, C-circle levels, APB levels, T-SCEs, and telomere synthesis in G2 and M). However, the authors also show that EXD2 KO is associated with shorter telomere length, suggesting that EXD2 is required for suppressing excessive ALT phenotypes that are associated with non-productive telomere synthesis (presumably this results in extratelomeric DNA consistent with the observation of enhanced C-circles). Importantly, and perhaps in contradiction to the observation of shorter telomeres the authors claim that EXD2 KO cells also have elevated levels of conservative telomere synthesis which is thought to be "productive" (telomere lengthening), and is measured by a modified co-FISH technique.

The authors go on to establish the genetic dependencies of many of their ALT phenotypes and find that many ALT phenotypes are only increased by EXD2 depletion when SMARCAL and MUS81 are present. Additionally, they find that RAD52 co-depletion does not alter the induction of ALT phenotypes observed in EXD2 KO cells, which they conclude is because EXD2 suppresses RAD52 independent ALT. Interestingly, they also find that TIFs are unresponsive to RAD51 inhibition, suggesting, perhaps, that EXD2 also suppresses RAD51 independent ALT.

Overall, the work is of good quality and makes an incremental contribution to the field. However, there were a few "major" concerns that should be addressed prior to publication:

- 1) The apparent discrepancy between telomere length and productive telomere synthesis needs to be addressed either by way of sufficient explanation as to why this is not actually a contradiction or with experimental data.
- 2) The role of RAD51 in all this needs to be clarified- it would seem that the results from this manuscript might suggest a RAD52/RAD51 independent pathway. Could a double KD be done to address this? Or is there another explanation?
- 3) In order to measure conservative telomere synthesis the authors use a modified co-FISH technique. However, the interpretation of different staining patterns produced by this technique, which is quite important to the paper, is extremely complicated and not clearly explained. If these results are to be believed the authors must make it extremely clear what exactly is being scored, and how different staining patterns differ from one another.

Additionally, there are many more minor issues with the paper that should be addressed:

- Line 28-29 in abstract the authors claim that RAD52-independent BIR relies specifically on intrachromosomal homology searches- is there evidence for this claim?
- Line 31-32 is poorly written. I believe the authors are trying to say that the initial processing of stalled forks dictates whether RAD52 dependent or independent BIR will be initiated, but this is somewhat unclear.
- Line 46-47 says "a telomere-enforced proliferation barrier is one of the first mechanisms limiting tumourigenesis." A review is cited as evidence. However, it seems very unlikely that this is true. In particular I take issue with the claim that it is one of the first mechanisms and I'm not even sure how this could be measured against, for instance, the contribution of checkpoint signaling.
- Line 84 is unclear. Is there a distinction being made between "recombination" and "BIR"? Or are they just saying that normally there is a mixture of RAD51/RAD52 mediated BIR and kd of EXD2

causes a shift to all RAD51 mediated BIR? Rewrite this more clearly.

- In Figure 1A there is a tiny misplaced white arrow across the "D" in "EXD" in the image label. Also, the label on the images are not centred in many cases.
- Across Fig 1. the individual reps should be shown with dots on the bar graphs. Additionally, it would be useful to see the distribution of colocalizations across individual cells for all your quants. Is there a subpopulation of cells with many colocalizations or are the ranges quite narrow? If you have some cells with many colocalizations this might indicate some cell cycle specificity to the function of EXD2.
- Overexpressed proteins can sometimes form erroneous nuclear condensates/foci if the levels of expression are too high. Expression of GFP-EXD2 construct should be shown via western blot, ideally with endogenous antibody so that the relative expression of the endogenous and GFP tagged proteins can be compared.
- Colocalization between GFP-EXD2 and PML alone (without a marker of the telomere), is somewhat meaningless. PML bodies do have functions outside of APBs so there is no way to infer that a protein localizing to a PML body has anything to do with ALT. The authors should exclude Fig 1D and instead quantify Fig 1E (APBs).
- Does EXD2 colocalize to tel+ telomeres?
- Line 108-109 uses the phrase "stability of telomeric DNA." I think more appropriate nomenclature for what is being measured would be "telomere protection", "telomere capping", or "formation of Telomere Dysfunction Induced Foci (TIF)."
- Markers of ALT including TIFs, c-circles, and APBs have all been shown to be cell cycle dependent. Additionally, telomere length as measured by FISH is not stable across the cell cycle (there isn't a doubling in telomere number in G1 vs G2 cells suggesting that many telomere foci in S/G2 actually represent a cluster of telomeres from sister chromatids). Same with RPA colocalization to telomeres. Therefore, the cell cycle must be considered to interpret much of this work. This could be addressed by evaluating the effect of EXD2 genetic manipulation on cell cycle phase (if a change does occur then ALT markers/telomere length would need to be assessed again in specific cell cycle phases).
- Please show EXD2 KO western blot confirmation in supplementary data.
- KO cell clones can result in adaptation as well as clonal variation that make interpretation of data from two clones quite difficult. Changes in cellular phenotypes could easily be an indirect outcome of the KO + selection process or simply random clonal variation. Fortunately, the authors address this by performing transient 3'UTR knock downs of EXD2 + rescue with either wt or exonuclease dead EXD2. This result is essential to the interpretation of the worked and should be moved into the main figure. If spacing is an issue Figure 2B does not need to be in the primary figure and can be moved to supplementary as PLA adds little to the overall interpretation.
- In figure 2C the authors use FISH to assess single chromatid signal free ends. They describe these events as "collapsed replication forks." However, I don't think it can necessarily be assumed that these events represent collapsed replication forks. If there is evidence for this, please cite it, otherwise please describe them more generally as "unreplicated telomeres." Also- what is being scored and the interpretation could be clearer in general. (lines 120-121).
- Line 128 and elsewhere refers to "T-DSB". This should be Telomere Dysfunction Induced Foci (TIF).
- Please present supp fig 3C with statistical analysis.

- The term “epistatic” is over used. The authors rely on this description as a crutch for when a result involving two genetic alterations cannot be readily explained mechanistically.
- Figure legend is incorrect for Sup Fig 1 (F and G).

Manuscript NCOMMS-22-12191-T

We would like to thank all reviewers for constructive appraisal of our manuscript that enabled us to further strengthen the key messages. We have performed additional experiments requested by the reviewers, including (i) knockdowns and phenotypic assays with additional siRNAs, (ii) analysis of some key phenotypes in additional ALT cell line, (iii) re-complementation analysis utilising WT as well as a nuclease-dead version of EXD2, (iv) additional experiments to analyse the engagement of RAD52-independent BIR in the absence of EXD2, and finally (v) analysis of recruitment of various factors to telomeres in WT as well as EXD2-deficient cells. We believe that we have thoroughly addressed the comments/suggestions raised. As a result, we feel that the manuscript has been substantially improved and is now suitable for consideration at Nature Communications.

Point-by-point responses:

Reviewer #1:

In the submitted manuscript (ms), Broderick et al reported that EXD2 localizes to telomeres in the ALT cells. Deficient EXD2 cells (using either siRNA targeting EXD2 and/or two genetic EXD2 knockout clones) manifested many dysfunctions/defects of ALT telomeres. Most important, they found that EXD2 might affect the sub-pathway selection in the ALT pathway. Unfortunately, many of the data presented in this ms are of poor quality. Therefore, it is very difficult to evaluate the potential important findings claimed by the authors.

Our reply: We thank the reviewer for recognising the importance of the study we have presented. We have addressed their specific comments to improve the quality of the data as outlined below.

“Specific points on the data quality in different figures: In general, many of the Western Blots (WB) are of poor quality. Some of the loading controls fluctuate too much (e.g., S1b, S1f, S2d etc).” Therefore, it is very difficult to judge the changes or lack of changes of protein of interest.

Our reply: We have repeated western blots to improve their quality throughout the manuscript.

The major findings also need to be repeated in another ALT cell line.

Our reply: We thank the reviewer for their comment and acknowledge that testing of key phenotypes in other cell lines is an important way to validate the phenotypes observed in our *EXD2*^{-/-} U2OS cells. Indeed, we had already validated the induction of C-circles in the absence of EXD2 in a panel of ALT-Reliant cell lines (**Fig. 3b**). We have also now included analysis of the key phenotypes associated with the loss of EXD2 utilising the IMR90 ALT-reliant cell line, namely the incidence of DNA breaks at the telomere (**New Supplementary Fig. 1d**), the incidence of ALT-Associated PML bodies (**New Supplementary Fig. 2b**) as well as the frequency of EdU incorporation at telomeres in isolated synchronised prometaphases and its dependence on RAD52 (**New Supplementary Fig. 3b**). In support of our initial data set, we observe an increased incidence of TIF, APBs and EdU incorporation in isolated synchronised prometaphase cells which is insensitive to RAD52 inhibition, thus phenocopying what we have reported utilising the *EXD2*^{-/-} U2OS cells.

Fig 1: The major weakness of the EXD2 telomere localization is that it lacks data using the antibody (Ab) recognizing the endogenous EXD2. If the EXD2 Ab is not suited for the immunofluorescence assay, the authors need to try other assays to demonstrate its recruitment to ALT telomeres. Since the authors rely heavily on the tagged EXD2 (GFP-EXD2 and FLAG-EXD2), they need to first demonstrate the expression level of these tagged proteins relative to the endogenous EXD2. The authors did attempt to address this for the FLAG-EXD2 in Fig S1b. However, the level for the endogenous EXD2 is too weak to be used to estimate the relative level of the exogenous FLAG-EXD2. The authors also need to quantify them. If the level is more than 10-fold, they should use viral vector to bring down the expression level of the tagged proteins to the endogenous EXD2. They also need to do the same for the GFP-EXD2. Secondly, they need to use the correct controls. For example, the appropriate control cell line for the GFP-EXD2 should be U2OS expressing GFP only rather than just the parental U2OS.

Our reply: We thank the reviewer for raising these points. To address this, we have repeated the analysis of GFP-EXD2 co-localisation to TRF2 and APBs using a control cell line expressing GFP only, and include a western blot showing that the expression of GFP-EXD2 is close to WT levels as measured by densitometry (**Rebuttal Fig. 1a**) in our model cell line (**New Fig. 1a-c and New Supplementary Fig. 1a**). We also note that the expression of GFP-EXD2 in our stable cell lines was also previously described^{1,2}.

As pointed out by the reviewer the expression of FLAG-EXD2 is much greater than the endogenous EXD2, however the level of over-expression between WT and the ND construct are similar (**Rebuttal Fig. 1b**). Crucially, only the WT-overexpressing cells rescue key phenotypes associated with EXD2-loss i.e. DNA breaks at the telomere (**New Fig. 2c**) while the nuclease-dead mutant expressing cells do not. This is in line with our previous analysis, showing that only the WT but not the ND mutant FLAG EXD2 rescues defective DNA end resection and sensitivity to a DSB-inducing agent¹. Furthermore, we have also generated stable pools of *EXD2*^{-/-} U2OS cells expressing WT and nuclease dead (ND) EXD2 and show that only WT cells but not the ND rescue the excess conservative telomere synthesis associated with EXD2 loss; note the similar levels of EXD2 overexpression of WT and ND variants (**Rebuttal Fig. 1b**). In addition, we also show that RAD52 inhibition reduces EdU incorporation in WT but not ND expressing cells (**New Supplementary Fig. 3c**). We therefore conclude that FLAG-EXD2 expression level does not affect the ability to re-complement key phenotypes associated with the loss of EXD2 or its nuclease activity.

Fig 2: The difference between wild-type (WT) U2OS and the two EXD2 knockout clones (EXD2^{-/-}) are very subtle in some of the experiments (2b, 2c, 2d). Therefore, all the data using the EXD2^{-/-} need to be rescued using the WT-EXD2.

Our reply: We have now analysed key phenotypes associated with EXD2 loss in EXD2-deficient cells re-complemented with either WT or nuclease dead (ND) mutant FLAG-EXD2 i.e. DNA breaks at the telomere or suppression of RAD52-independent BIR (**New Fig. 2c and New Supplementary Fig. 3c**). These analyses clearly show that the nuclease activity of EXD2 is required in both cases. Moreover, the significance of differences between WT and EXD2-deficient cells for various phenotypes are determined by appropriate statistical analyses in all cases.

Fig 3: The difference between wild-type (WT) U2OS and the two EXD2 knockout clones (EXD2^{-/-}) are very subtle in some of the experiments (3a, 3b, 3d). Therefore, all the data using the EXD2^{-/-}

/- need to be rescued using the WT-EXD2. In addition, the figures 3e and 3f and their legends are not matching. The author need perform the TRF assay to quantify the telomere length.

Our reply: We agree with the reviewer that an independent measurement of telomere length is a good validation of the quantitative FISH data we had already included. Therefore, in collaboration with Genomic Vision, we have used a highly sensitive DNA-combing based approach to measure the average telomere lengths in both U2OS WT as well as *EXD2*^{-/-} cells. The new analysis supports our initial findings showing a significantly shorter average telomere length in *EXD2*^{-/-} cells as compared to WT (**New Fig. 3g**). We have also corrected the legends and these are now matching in all cases.

Fig 4: The difference between wild-type (WT) U2OS and the two EXD2 knockout clones (EXD2^{-/-}) are very subtle in some of the experiments (4a to 4c). Therefore, all the data using the EXD2^{-/-} need to be rescued using the WT-EXD2. The results between siRad52 and Rad52i are inconsistent for cl.1. Why is that?

Our reply: We respectfully disagree with the reviewer; all phenotypes shown are accompanied by appropriate statistical analyses carried out to determine effects in multiple biological repeats. The trends between siRAD52 and RAD52i are consistent between the WT U2OS cells and two independently derived clones of U2OS *EXD2*^{-/-}. In both cases, depletion or inhibition of RAD52 does not reduce the rates of EdU incorporation in *EXD2*^{-/-} cells, while in the WT cells loss or inhibition of RAD52 leads to reduced levels of EdU incorporation at the telomere as determined by statistical analysis.

Fig 5: Are there cell cycle and/or cell death changes when co-depleted with siBLM, siSLX4, siMUS82 and siSMARCAL1?

Our reply: As shown in the previous version of the manuscript, only depletion of BLM results in a synthetic sick phenotype upon *EXD2* loss (**Old Fig. 6a, now New Fig. 6a**). Depletion of SLX4, MUS81 or SMARCAL1 were also shown in the previous version of the manuscript to have no synthetic sick phenotype with *EXD2* loss as assayed by colony formation assay (**now New Fig. 6a, New Supplementary Fig. 8d, New Supplementary Fig. 9f**).

Nevertheless, we have also analysed the cell cycle profiles, by FACS, of WT and *EXD2*^{-/-} U2OS cells upon depletion of various proteins by siRNA, as requested by the reviewer. This data is presented in the **Rebuttal Fig. 2a and b**. We observe that depletion of SMARCAL1 and MUS81 caused a slight increase of G2/M cells in both WT and *EXD2*-deficient cells while depletion of SLX4 had no discernible effect on cell cycle for both WT and *EXD2*-deficient cells. BLM-depletion lead to an increase in G1 and a decrease in S-phase cells for both WT and *EXD2*-deficient cells. Interestingly, and in support of the colony formation data showing a synthetic sick relationship between *EXD2* and BLM (**New Fig. 6a**) depletion of BLM in *EXD2*-deficient cells leads to an increased sub-G1 population as analysed by FACS compared to WT cells depleted for BLM, which may indicate increased apoptosis/necrosis (i.e. cell death) under these conditions (**Rebuttal Fig. 2b**).

Fig 6: Both EXD2^{-/-} clones seem to be quick sick on their own (Figs 6, S4, S5). Are there any cell cycle and growth rate changes by themselves? In addition, all the experiments with positive results using siRNA need to be repeated with the second independent siRNA.

Our reply: As requested by the reviewer we analysed proliferation of *EXD2*-deficient U2OS cells under normal growth conditions. Our data show that *EXD2*-deficient cells exhibit a mild proliferative defect compared to WT control cells (**Rebuttal Fig. 3a**); which is also consistent with

our DNA fiber analysis (**New Supplementary Fig. 1f**). Crucially, for all colony formation assays depicted in the manuscript, the relative fraction of colonies is calculated, whereby the colonies formed on the control plate (U2OS WT or EXD2-deficient cells treated with control siRNA) is used to normalise for each cell line, and the relative surviving fraction calculated from this.

As requested by the reviewer we also carried out cell cycle analyses for WT and EXD2-deficient U2OS cells. Here, we do not observe a significant difference in the relative fractions of G1, S or G2/M cells (**Rebuttal Fig. 3b**).

As requested by the reviewer, we have also performed additional analyses using independent siRNAs for Mus81, SMARCAL1, BLM, DNA and Pold3. These new data show that depletion of MUS81 or SMARCAL1 using independent siRNAs has the same effect i.e. reduces the incidence of DSBs at the telomere as well as EdU incorporation at the telomere in EXD2-deficient cells (**New Supplementary Fig. 1i and 7d**). Analysis of synthetic sick interactions between EXD2 loss and depletion of BLM, DNA2 and Pold3 was confirmed using independent siRNAs (**New Supplementary Fig. 9a, c and e**).

Fig S1: The WB for SMARCAL1 is of poor quality.

Our reply: This has been repeated and a better quality western included (**New Supplementary Fig. 1h**).

Fig S2: The WB for Rad52 is of poor quality.

Our reply: This has been repeated and a better quality western included (**New Supplementary Fig. 3a**).

Fig S4: The WB for SLX4 is of poor quality. SLX4 seems to be drastically reduced in the EXD2-/- cells. Why is that?

Our reply: We apologize for the quality of the blots. As requested by the reviewer we repeated the western blot for SLX4 and a better-quality blot is now included (**New Supplementary Fig 7b**). We do not observe a large difference in the expression of SLX4 in EXD2-deficient cells. Nevertheless, to rule out that EXD2-loss results in deficient SLX4 localisation/function in ALT we have also analysed the association of SLX4 to TRF2 (a surrogate marker of the telomere) as well as to APBs by four colour immunofluorescence staining (**New Supplementary Fig. 10a**). These analyses show increased SLX4 association with telomeres in EXD2-deficient cells, consistent with increased DNA breaks and recombination events at telomeres in EXD2-deficient U2OS cells.

Fig S5: Human DNA2 has 1060 aa. The size in the WB (above 170 KDa) seems to be very high relative to its predicated size (5a). Furthermore, previous publications showed that both recombinant and endogenous DNA2 runs around 130 KDa on a SD-PAGE (PMID: 23604072 and 31216032). Please make sure that the DNA2 Ab used in the WB recognizes the right protein.

Our reply: This western blot was run on a 3-8% Tris-Acetate gel where migration of large molecular weight proteins can, in our experience, vary largely depending on the factor being analysed. We have repeated this western blot running the samples on a 4-12% Bis-Tris gel, for

both the original siRNA used in the paper, and for the second independent siRNA used to validate our colony formation phenotype. In both cases we observe knockdown of DNA2 with a band closer to the predicted molecular weight as highlighted by the reviewer (**New Supplementary Fig. 9b and c**).

Minor points:

Line 43: Dagg et al (2017, PMID 28636942) showed that a small percentage of TLM negative tumors can survive >200 population doublings with ever-shorter telomeres. Therefore, the statement may not be completely correct.

Our reply: We respectfully disagree, the statement that "To achieve unlimited proliferative potential, cells must maintain their telomeres". We appreciate reviewer suggestion but, in our view, unlimited means "not limited or restricted in terms of number".

Line 48-50: Inappropriate citations. Recommended citations: PMID – 34069193, 266540511

Our reply: This citation has been changed.

Line 50-51: Imprecise statement. For example, prior to chemotherapy, the overall survival rate for osteosarcoma was a dismal 20%. With chemotherapy, the overall 5-year survival rate increase to 60%–70% (PMID: 21559216).

Our reply: This statement has been amended.

Line 301-304: Pan et al (2017, PMID: 28673972) first reported the involvement of FANCM in the ALT pathway. In the same study, they reported synthetic lethal interaction between FANCM and BLM.

Our reply: We apologize for this omission; this has been amended.

Reviewer #2:

The manuscript by Broderick et al investigated the role of EXD2, a nuclease that has been implicated in HR, fork restart and fork protection, in ALT cells. The authors showed that EXD2 is recruited to ALT telomeres and regulates ALT hallmarks. They went on to further characterize RAD52-independent telomere DNA synthesis in EXD2^{-/-} cells. They concluded that EXD2 dictates pathway choice in ALT cells.

Our reply: We thank the reviewer for recognising the value of the study and highlighting the key phenotypes that we have characterised.

Major issues:

1. *The authors fell short of demonstrating that EXD2 dictates the pathway choice at ALT telomeres. Although they showed that telomere DNA synthesis in EXD2^{-/-} cells is RAD52-independent, EXD2^{-/-} cells exhibit telomere shortening. The latter is consistent with the notion that RAD52-dependent BIR is impaired at ALT telomeres in EXD2^{-/-} cells. It is possible that EXD2 does not regulate the pathway choice per se but rather is engaged in RAD52-dependent BIR and that EXD2^{-/-} cells simply rely on RAD52-independent BIR for telomere synthesis because of impaired RAD52-dependent BIR.*

Our reply: We thank the reviewer for pointing out that the loss of EXD2 leads to RAD52-independent DNA synthesis at the telomere, and point out that the incidence of both EdU incorporation at telomeres EXD2-deficient U2OS cells and conservative telomere synthesis in asynchronous cells lacking EXD2 are both increased (**New Fig. 4a-c**), while the average telomere length is shorter as assayed by quantitative FISH and now by DNA combing (**New Fig. 3f and g**). We therefore propose that in the absence of EXD2, cells rely on a RAD52-independent BIR which may be less processive/efficient than RAD52-dependent synthesis, which would explain the elevated levels of telomeric synthesis but reduced overall telomere length observed in EXD2-deficient U2OS. We agree with the reviewer that it is possible that EXD2 promotes RAD52-dependent BIR and that in the absence of EXD2, cells rely on RAD52-independent BIR, which to our mind constitutes pathway “choice”. How we choose to refer to this mechanism may be an issue of semantics.

2. *The authors show that EXD2 loss leads to reduced levels of SLX4 (Fig. S4a), MUS81 (Fig. S4b), and MRE11 (Fig. S2d). Given that SLX4, MUS81 and MRE11 are known to be engaged in ALT, their finding raises a concern as to whether or what portion of the observed phenotypes in EXD2^{-/-} cells are due to EXD2 itself.*

Our reply: We thank the reviewer for this important comment. We have since provided better quality western blots upon testing the knockdown of SLX4 (**New Supplementary Fig 7b** and Mus81 (**New Supplementary Fig. 1i**) and importantly, do not observe a large difference in the levels of these proteins in EXD2-deficient cells. Nevertheless, to rule out the possibility that loss of EXD2 is impairing the recruitment/retention of these factors to telomeres and to APBs as part of their role in the ALT mechanism, we have analysed the association of Mus81, SLX4 and MRE11 to TRF2 (a surrogate marker of the telomere) and to APBs by four colour immunofluorescence staining (**New Supplementary Fig 8a, New Supplementary Fig. 10a and b**). These new data do not show defective localisation for any of these factors; with perhaps slightly elevated association of SLX4 and MUS81 to telomeres and APBs in EXD2-deficient cells, which would be consistent with elevated levels of stalled forks at telomeres requiring MUS81-

dependent cleavage and/or increased incidence of DNA recombination events at telomeres in EXD2-deficient U2OS cells.

Moreover, in the previous version of the manuscript we have also tested the co-depletion of EXD2 with both SLX4 and MUS81 and found that their depletion did not result in a synthetic sick phenotype, in contrast to depletion of BLM (**now New Fig. 6a, New Supplementary Fig.8d**).

Importantly, we have also re-complemented key phenotypes (DNA breaks at the telomere and EdU incorporation at the telomere as well as its dependence on RAD52) in EXD2-deficient cells expressing either WT or nuclease-dead EXD2 (**now New Fig. 2c and New Supplementary Fig. 3c**). These analyses clearly show that the nuclease activity of EXD2 is required in both cases, which strongly suggests that the observed phenotypes are not due to the mis-localisation/loss of other factors involved in the ALT mechanism.

3. Previously, Zhang et al (2019) reported that RAD51 and MRE11 inhibit RAD52- independent BIR-mediated C-circles. Does EXD2 loss affect the level of RAD51? Do RAD51 and MRE11 inhibit RAD52-independent BIR-mediated telomere DNA synthesis and C-circles in EXD2-/- ALT cells?

Our reply: We thank the reviewer for this important comment. We observe that EXD2 loss does not impact the expression levels of RAD51 in U2OS cells (**New Supplementary Fig. 4c**). We have also extended our analysis of C-circle abundance in EXD2-deficient cells to analyse both the impact of knockdown of MRE11 and RAD51 on this phenotype (**New Supplementary Fig. 4b and c**). These new analyses show that loss of EXD2 is associated with increased levels of C-circles, while the co-depletion of MRE11 or RAD51 does not result in a significant change in C-circle abundance.

4. The authors claimed that there is an epistatic relationship between EXD2, RAD51 and MRE11 in repair of T-DSBs in ALT and that RAD51- and MRE11-dependent repair pathways can be ruled out as the main repair pathways utilized in ALT cells in the absence of EXD2. This claim was based solely on the 53BP1/ALT telomere colocalization experiment and therefore it is not sufficiently supported. The authors should at least investigate if EXD2 loss affects association of HR proteins with ALT telomeres and vice versa.

Our reply: As requested by the reviewer we have now performed additional analyses testing the association of various factors to telomeres/APBs in the absence of EXD2. These new data do not show defective localisation of RAD51, SMARCAL1, Mus81, SLX4 or MRE11 but show a decreased localisation of RAD52 consistent with increased RAD52-independent BIR being utilised in EXD2-deficient cells (**New Supplementary Fig. 6a and b, New Supplementary Fig. 7c, New Supplementary Fig. 8a and New Supplementary Fig. 10a-c**).

Moreover, inhibition of HR by addition of inhibitors targeting RAD51 (canonical HR) or RAD52 (BIR) does not affect GFP-EXD2 localisation to ALT telomeres (as measured by GFP-EXD2/TRF2 co-localisation) (**Rebuttal Fig. 4a and b**).

5. The authors claimed that EXD2 nuclease activity is required for the pathway choice, but they fell short of testing EXD2 nuclease-dead mutants in RAD52- independent BIR in ALT.

Our reply: We thank the reviewer for raising this point. We addressed this remark by generating pools of EXD2-deficient cells stably expressing WT or nuclease-dead mutant FLAG-EXD2 and by assaying the frequency of EdU incorporation at the telomere in isolated synchronised

prometaphase cells treated with DMSO or a RAD52-inhibitor. These new data show that cells expressing WT FLAG-EXD2 rescue the elevated levels of EdU incorporation at the telomere rendering this synthesis sensitive to RAD52-inhibition, while the nuclease-dead expressing cells fail to do so (**New Supplementary Fig. 3c**).

6. *The authors suggest that EXD2 loss leads to excess replication fork reversal (by SMARCAL1) at ALT telomeres, which are cleaved by MUS81 and repaired by RAD52-independent BIR. Does EXD2 loss lead to a reduction in RAD52 association with ALT telomeres? Does EXD2 loss lead to an increase in telomere association of SMARCAL1, MUS81 and POLD3?*

Our reply: As requested by the reviewer we have analysed the recruitment of several factors to ALT telomeres in EXD2-deficient vs proficient cells. These analyses show decreased association of RAD52 (consistent with the use of RAD52-independent telomere synthesis) with no effect on the limited association of RAD51 to telomeres in WT vs EXD2-deficient U2OS cells (**New Supplementary Fig. 6a and b**). Moreover, we have also analysed the co-localisation of SMARCAL1, Mus81 and POLD3 with TRF2 and APBs by four colour immunofluorescence analysis (**New Supplementary Fig. 7c, New Supplementary Fig. 8a and New Supplementary Fig 10c**). Here, we do not observe a defective localisation of these factors to telomeres. On the contrary, we observe a slight but significant increase in their association to both TRF2 and APBs, consistent with elevated levels of RAD52-independent recombination being carried out at telomeres in EXD2-deficient U2OS cells.

7. *In their model, the authors proposed that HR-mediated fork restart at ALT telomeres is disabled in the absence of EXD2 but they did not provide supporting data.*

Our reply: The assumption that fork-restart by canonical HR is impaired in EXD2-deficient cells is based on the analysis showing that depletion or inhibition of RAD51 or MRE11 does not lead to a further increase in breaks at the telomere in EXD2-deficient cells (**New Fig. 2f, New Supplementary Fig. 1g**). This data suggests that the canonical HR (RAD51-dependent) does not contribute to fork restart in ALT cells lacking EXD2. In line with this notion, we have previously shown that EXD2 is required for efficient canonical HR^{1,2}.

Furthermore, in the current manuscript we show that loss of EXD2 or its nuclease activity leads to the increased incidence of DNA breaks at the telomere of ALT-reliant U2OS cells (**New Fig. 2a, b and c**) as well as impaired DNA replication as assayed by the DNA fiber technique in both, ALT-reliant (**New Supplementary Fig. 1f**) and non-ALT cells². Crucially, the nuclease activity of EXD2 and fork regression mediated by SMARCAL1 is required to suppress DNA breaks associated with EXD2-loss at the telomere in ALT-reliant cells (**New Fig. 2c and g; New Supplementary Fig. 1i**). Based on these analyses we propose a model whereby EXD2 likely processes the regressed forks that are remodelled by SMARCAL1 during replication stress to prevent fork collapse and allow for efficient fork restart. In the absence of EXD2 perturbed replication forks will collapse (a process likely mediated by MUS81 in ALT-reliant U2OS (**New Supplementary Fig. 1i, New Supplementary Fig. 8c**)). We also propose that at the telomere in ALT-reliant cells these collapsed forks engage a RAD52-independent BIR mechanism.

Minor issues:

1. *Ref 16 on line 67 is incorrectly cited.*

Our reply: This has been amended.

2. Line 171, “than” is misspelled.

Our reply: This has been corrected.

3. Line 161, Fig. 2d is incorrectly cited.

Our reply: This has been now amended.

4. Fig. S4a, the SLX4 blot is of poor quality. SLX4 does not appear to be knocked down in the last two lanes.

Our reply: We apologise for the poor quality of the blots. We have re-run these westerns and a new, better quality blots is now included that confirms SLX4 knockdown (**New Supplementary Fig, 7b**).

Reviewer #3:

The authors describe the function of EXD2 in the ALT mechanism and attempt to differentiate its relative contributions to RAD52-dependent and RAD52-independent telomere synthesis. The authors found that EXD2 colocalizes with ALT telomeres and APBs. Additionally, they thoroughly demonstrate that EXD2 KO cells have enhanced ALT phenotypes (TIF formation, RPA colocalization to telomeres, C-circle levels, APB levels, T-SCEs, and telomere synthesis in G2 and M). However, the authors also show that EXD2 KO is associated with shorter telomere length, suggesting that EXD2 is required for suppressing excessive ALT phenotypes that are associated with non-productive telomere synthesis (presumably this results in extratelomeric DNA consistent with the observation of enhanced C-circles). Importantly, and perhaps in contradiction to the observation of shorter telomeres the authors claim that EXD2 KO cells also have elevated levels of conservative telomere synthesis which is thought to be “productive” (telomere lengthening), and is measured by a modified co-FISH technique.

Our reply: We thank the reviewer for recognising the importance of our work. We note that our data is not inconsistent with a model whereby there is elevated engagement in RAD52-independent BIR (as evidenced by the elevated levels of EdU incorporation observed in G2 and prometaphase cells as well as the increased incidence of conservative telomere synthesis observed in asynchronous cells by 2-replication round CO-FISH (**New Fig. 4a-c**)) whereby this RAD52-independent synthesis is less productive/efficient than the RAD52-dependent mechanism normally utilised by ALT-reliant cells, resulting in reduced average telomere length in the EXD2-deficient U2OS compared to the parental cells (**New Fig. 3g**).

The authors go on to establish the genetic dependencies of many of their ALT phenotypes and find that many ALT phenotypes are only increased by EXD2 depletion when SMARCAL and MUS81 are present. Additionally, they find that RAD52 co-depletion does not alter the induction of ALT phenotypes observed in EXD2 KO cells, which they conclude is because EXD2 suppresses RAD52 independent ALT. Interestingly, they also find that TIFs are unresponsive to RAD51 inhibition, suggesting, perhaps, that EXD2 also suppresses RAD51 independent ALT. Overall, the work is of good quality and makes an incremental contribution to the field. However, there were a few “major” concerns that should be addressed prior to publication: 1) The apparent discrepancy between telomere length and productive telomere synthesis needs to be addressed either by way of sufficient explanation as to why this is not actually a contradiction or with experimental data.

Our reply: We believe that the engagement of RAD52-independent BIR may be less processive/efficient than RAD52-dependent BIR, explaining the elevated levels of telomere synthesis (**New Fig. 4a-c**) but overall shorter telomere length (**New Fig. 3f and g**).

2) The role of RAD51 in all this needs to be clarified- it would seem that the results from this manuscript might suggest a RAD52/RAD51 independent pathway. Could a double KD be done to address this? Or is there another explanation?

Our reply: We thank the reviewer for this important comment. To further characterise the role of RAD51 in the RAD52-independent BIR carried out in EXD2-deficient cells we have analysed the localisation of RAD51 to ALT telomeres in both WT and EXD2-deficient U2OS (**New**

Supplementary Fig. 6b). This new data set shows limited association of RAD51 with telomeres in both WT and *EXD2*^{-/-} cells.

Moreover, we have also analysed the frequency of EdU incorporation in isolated synchronised prometaphase cells in both WT and *EXD2*^{-/-} U2OS treated with a RAD51 inhibitor (B-02) alone or co-treated with both a RAD52 inhibitor (AICAR) and B-02 (**New Supplementary Fig. 3d**). Here, we observe that inhibition of RAD51 had no effect on the frequency of EdU incorporation in either WT or *EXD2*-deficient cells while further addition of the RAD52 inhibitor AICAR lead to reduced levels of EdU incorporation in WT but not *EXD2*-deficient cells and as such, phenocopies treatment with AICAR alone (**New Fig. 4b**); (of note, single treatment with AICAR was included in the previous version of the manuscript (**old Fig. 4b**)). Altogether, these new data suggest that RAD51 does not seem to have a role in promoting ALT telomere synthesis in the absence of *EXD2*.

In order to measure conservative telomere synthesis the authors use a modified co-FISH technique. However, the interpretation of different staining patterns produced by this technique, which is quite important to the paper, is extremely complicated and not clearly explained. If these results are to be believed the authors must make it extremely clear what exactly is being scored, and how different staining patterns differ from one another.

Our reply: We apologise for the confusion. We have now added an additional description in the main manuscript text clarifying exactly which depicted scenarios are scored. This information and a schematic representation of the scenarios scored as conservative events are also present in the figure and figure legend (**New Supplementary Fig. 5b**).

Minor issues:

Line 28-29 in abstract the authors claim that RAD52-independent BIR relies specifically on intrachromosomal homology searches- is there evidence for this claim?

Our reply: We think it most likely that RAD52-independent BIR is carried out intrachromosomally due to the proximity of the sister telomere, however we cannot formally rule out that interchromosomal synthesis or synthesis utilising a template on the same chromosome arm within the telomere repeat array is used. We have therefore amended this statement in the new version of the manuscript.

Line 31-32 is poorly written. I believe the authors are trying to say that the initial processing of stalled forks dictates whether RAD52 dependent or independent BIR will be initiated, but this is somewhat unclear.

Our reply: This has been now rewritten in the new version of the manuscript.

Line 46-47 says “a telomere-enforced proliferation barrier is one of the first mechanisms limiting tumourigenesis.” A review is cited as evidence. However, it seems very unlikely that this is true. In particular I take issue with the claim that it is one of the first mechanisms and I’m not even sure how this could be measured against, for instance, the contribution of checkpoint signaling.

Our reply: As requested by the reviewer this has been removed.

Line 84 is unclear. Is there a distinction being made between “recombination” and “BIR”? Or are they just saying that normally there is a mixture of RAD51/RAD52 mediated BIR and kd of EXD2 causes a shift to all RAD51 mediated BIR? Rewrite this more clearly.

Our reply: This has been now clarified within the text.

In Figure 1A there is a tiny misplaced white arrow across the “D” in “EXD” in the image label. Also, the label on the images are not centred in many cases.

Our reply: This has been amended.

Across Fig 1. The individual reps should be shown with dots on the bar graphs. Additionally, it would be useful to see the distribution of colocalizations across individual cells for all your quants. Is there a subpopulation of cells with many colocalizations or are the ranges quite narrow? If you have some cells with many colocalizations this might indicate some cell cycle specificity to the function of EXD2.

Our reply: This has been amended with individual replicas now shown. Moreover, we present a quantification of co-localisations of GFP-EXD2 with TRF2 and APBs in U2OS cells stably expressing GFP-EXD2 (**Rebuttal Fig. 5a and b**) which show a relatively narrow distribution of numbers of co-localisations for both TRF2 and APBs with many cells having no co-localisations. This suggests that EXD2 association to telomeres/APBs could be cell cycle regulated. Interestingly, the distribution of localisations to TRF2 and APBs is very similar, suggesting that GFP-EXD2 mostly associates with APBs which may be of interest to study further.

Overexpressed proteins can sometimes form erroneous nuclear condensates/foci if the levels of expression are too high. Expression of GFP-EXD2 construct should be shown via western blot, ideally with endogenous antibody so that the relative expression of the endogenous and GFP tagged proteins can be compared.

Our reply: This has now been included (**New Supplementary Fig. 1a**).

Colocalization between GFP-EXD2 and PML alone (without a marker of the telomere), is somewhat meaningless. PML bodies do have functions outside of APBs so there is no way to infer that a protein localizing to a PML body has anything to do with ALT. The authors should exclude Fig 1D and instead quantify Fig 1E (APBs).

Our reply: We agree with the reviewer comment and **New Fig. 1a-c** now includes analysis of TRF2 and APB localisation.

Does EXD2 colocalize to tel+ telomeres?

Our reply: Analysis of localisation of FLAG-EXD2 to telomeres by proximity ligation assay using antibodies specific to FLAG and TRF2 (a surrogate marker of the telomere) in cells stably expressing FLAG-EXD2^{1,2} show relatively modest association of FLAG-EXD2 with TRF2 in HeLa cells, with significantly greater association in U2OS cells expressing FLAG-EXD2 (**Rebuttal Fig.**

6a and b), suggesting that EXD2 may associate with telomeres in non-ALT cells, which likely reflects its role in promoting replication fork stability throughout the genome².

Line 108-109 uses the phrase “stability of telomeric DNA.” I think more appropriate nomenclature for what is being measured would be “telomere protection”, “telomere capping”, or “formation of Telomere Dysfunction Induced Foci (TIF)”.

Our reply: This has been rephrased.

Markers of ALT including TIFs, c-circles, and APBs have all been shown to be cell cycle dependent. Additionally, telomere length as measured by FISH is not stable across the cell cycle (there isn't a doubling in telomere number in G1 vs G2 cells suggesting that many telomere foci in S/G2 actually represent a cluster of telomeres from sister chromatids). Same with RPA colocalization to telomeres. Therefore, the cell cycle must be considered to interpret much of this work. This could be addressed by evaluating the effect of EXD2 genetic manipulation on cell cycle phase (if a change does occur then ALT markers/telomere length would need to be assessed again in specific cell cycle phases).

Our reply: We have carried out an analysis of the cell cycle profile of WT and EXD2-deficient cells (**Rebuttal Fig.3b**) which shows no significant differences in cell cycle profiles. This, to our minds, negates any potential problems when analysing phenotypes in asynchronous cells (TIFs APBs, etc). For telomere length measurement by quantitative FISH, these are carried out on metaphase spreads and are thus not affected by any differences in cell cycle profile of cycling cells. Importantly, we have now independently verified telomere length using a DNA combing-based approach, which shows the same phenotype (**New Fig. 3g**).

Please show EXD2 KO western blot confirmation in supplementary data.

Our reply: This has now been included (**New supplementary Fig. 1b**).

KO cell clones can result in adaptation as well as clonal variation that make interpretation of data from two clones quite difficult. Changes in cellular phenotypes could easily be an indirect outcome of the KO + selection process or simply random clonal variation. Fortunately, the authors address this by performing transient 3'UTR knock downs of EXD2 + rescue with either wt or exonuclease dead EXD2. This result is essential to the interpretation of the worked and should be moved into the main figure. If spacing is an issue Figure 2B does not need to be in the primary figure and can be moved to supplementary as PLA adds little to the overall interpretation.

Our reply: This has been added to the main figures (**New Fig. 2c**). We have also performed re-complementation analysis using EXD2-deficient cells expressing either WT or nuclease-dead EXD2, analysing EdU incorporation at the telomere in isolated synchronised prometaphase cells (**New Supplementary Fig. 3c**). These analyses show that the nuclease activity of EXD2 is required to suppress excess EdU incorporation.

In figure 2C the authors use FISH to assess single chromatid signal free ends. They describe these events as “collapsed replication forks.” However, I don't think it can necessarily be assumed

that these events represent collapsed replication forks. If there is evidence for this, please cite it, otherwise please describe them more generally as “unreplicated telomeres.” Also- what is being scored and the interpretation could be clearer in general. (lines 120-121).

Our reply: This has been changed.

Line 128 and elsewhere refers to “T-DSB”. This should be Telomere Dysfunction Induced Foci (TIF).

Our reply: This has been changed throughout the manuscript where appropriate.

Please present supp fig 3C with statistical analysis.

Our reply: The analysis in Supplementary Fig. 3c was performed as carried out previously by the Shay lab³. When comparing the percentage of conservative events observed between WT and EXD2-deficient cells we observe a slight increase in the percentage of conservative events in EXD2-deficient cells compared to WT which is not statistically significant. However, this experimental setup is insufficient to quantify the frequency of conservative synthesis events, which we determine by 2-replication round CO-FISH (**New Fig. 4c**) and by proxy using EdU incorporation at the telomere in synchronised G2 and prometaphase cells (**New Fig. 4a and b**) where we observe a significant increase in EXD2-deficient cells compared to WT.

The term “epistatic” is over used. The authors rely on this description as a crutch for when a result involving two genetic alterations cannot be readily explained mechanistically.

Our reply: This has been rephrased.

Figure legend is incorrect for Sup Fig 1 (F and G).

Our reply: This has been amended.

References

- 1 Broderick, R. *et al.* EXD2 promotes homologous recombination by facilitating DNA end resection. *Nature cell biology* **18**, 271-280, doi:10.1038/ncb3303 (2016).
- 2 Nieminuszczy, J. *et al.* EXD2 Protects Stressed Replication Forks and Is Required for Cell Viability in the Absence of BRCA1/2. *Molecular cell*, doi:10.1016/j.molcel.2019.05.026 (2019).
- 3 Min, J., Wright, W. E. & Shay, J. W. Alternative Lengthening of Telomeres Mediated by Mitotic DNA Synthesis Engages Break-Induced Replication Processes. *Molecular and cellular biology* **37**, e00226-00217, doi:10.1128/mcb.00226-17 (2017).

Rebuttal Figure 1: Quantification of FLAG-EXD2 expression.

a) Densitometry analysis of western blot from New Supplementary Fig. 1a calculating relative expression levels of GFP-EXD2 vs endogenous EXD2. All densitometry signals were normalized to Tubulin and fold expression relative to endogenous EXD2 calculated.

b) Densitometry analysis of western blot from New Supplementary Fig. 1e calculating relative overexpression levels of WT and nuclease dead (ND) mutant FLAG-EXD2 in U2OS cells. All densitometry signals were normalized to Tubulin and fold overexpression relative to endogenous EXD2 calculated.

Rebuttal Figure 2: Analysis of BLM, SLX4, MUS81 and SMARCAL1 depletion on cell cycle profile.

a) Histogram plots of gated single cells for FACS analysis of propidium iodide (PI) staining of WT U2OS and EXD2-deficient cells treated with either control siRNA (si Luciferase) or siRNA targeting BLM, SLX4, MUS81 or SMARCAL1, as indicated. Percentage of cells within single cell population in G1, S or G2/M of the cell cycle is presented in each case.

b) Histogram plot of ungated FACS analysis of propidium iodide (PI) staining of WT U2OS and EXD2-deficient cells treated with siRNA targeting BLM. Sub-G1 population is indicated with a vertical line.

Rebuttal Figure 3: Proliferation and cell cycle analysis of EXD2-deficient U2OS cells.

a) Proliferation analysis of U2OS WT and EXD2-deficient clones by CellTiter-Glo analysis. Proliferation values were normalized to day 0 measurements and fold change in proliferation calculated (n= 3 independent experiments, bars represent +/- SEM).

b) Quantification of FACS analysis of BrdU/PI staining of WT U2OS and EXD2-deficient cells. Cells were labelled for 30 min with 10 μ M BrdU followed by harvesting for FACS analysis and staining using an antibody specific to BrdU under denaturing conditions followed by PI staining. The percentage of cells in G1, S or G2/M are provided as well as representative images of FACS plots (BrdU vs PI) and gating strategy (n= 3 independent experiments, bars represent +/- SEM, statistical significance was determined by Student's t-test).

Rebuttal Figure 4: Inhibition of HR does not abrogate GFP-EXD2 localisation to telomeres.

a) and **b)** Quantification of GFP/TRF2 co-localisations (percentage cells with co-localisations and average number of co-localisations per positive cell) in U2OS cells stably expressing GFP or GFP-EXD2 as indicated. Cells were treated with RAD51 inhibitor (B-02, 25 μ M) or RAD52 inhibitor (AICAR, 40 μ M) for 2h as indicated. n= 150 cells from 3 independent experiments, statistical significance determined by Student's t-test, bars represent mean +/- SEM.

Rebuttal Figure 5: Analysis of GFP-EXD2 co-localisation with TRF2 and APBs.

a-b) Quantification of the numbers of GFP-EXD2 co-localisations with TRF2 (panel a) and APBs (panel b) in individual cells from New Fig. 1a-c (n= 150 cells from 3 independent experiments, bars represent mean +/- SEM).

Rebuttal Figure 6: EXD2 associates with non-ALT telomeres.

a-b) FLAG-EXD2 localises to telomeres in both non-ALT and ALT cells as assayed by its association with TRF2 (Shelterin component) using the PLA assay. The number of PLA foci per cell is quantified in Non-ALT reliant HeLa cells (negative control) and HeLa EXD2^{-/-} cells stably expressing FLAG-EXD2 WT along with U2OS cells (an additional negative control) or cells stably expressing FLAG-EXD2 which acts as a positive control. PLA signal appears in red, DAPI acts as a nuclear stain (n= at least 220 cells from 2 independent experiments, statistical analysis was carried out by Mann-Witney test, error bars represent +/- SEM, scale bar = 10 μ m).

REVIEWER COMMENTS

Reviewer #1 (Remarks to the Author):

The quality of the data in the revised manuscript by Broderick and colleagues has been dramatically improved. However, there are still a few major issues needed to be addressed before I could recommend for its acceptance.

Major points:

1. The changes shown in Fig 2B and Fig S1C- are quite mild. γ H2AX should be a better DNA damage marker, which can better capture the mixed forms of DNA breaks induced by fork collapse. In addition, the authors should also perform pChk1 staining, which would re-enforce their hypothesis that EXD2 deficiency causes replication problems at the ALT telomeres.
2. Telomeric SMARD analysis is a more direct method to address whether there is DNA replication defects at telomeres. The authors need to perform the SMARD assay using the EXD2^{-/-} cells.
3. For the C-circle blot in Fig 3A, visually, I am not convinced that there is any difference between U2OS and EXD2^{-/-} cl.1. Similarly, in Fig 3B, the C-circle increases in siEXD2 U2OS cells is also very mild. To confirm these very mild C-circle increases in the EXD2 deficient U2OS cells, both EXD2^{-/-} cl.1 and siEXD2 need to be rescued.

Minor points:

- Line-109: mis-spelled "Dysfunction"
- Please provide a reference for the validity of TeloSizer

Reviewer #2 (Remarks to the Author):

The authors thoughtfully addressed my comments and the manuscript is greatly strengthened and in good shape for publication in Nat Commun.

Reviewer #3 (Remarks to the Author):

The authors describe the function of EXD2 in the ALT mechanism and attempt to differentiate its relative contributions to RAD52-dependent and RAD52-independent telomere synthesis. The authors found that EXD2 colocalizes with ALT telomeres and APBs. Additionally, they thoroughly demonstrate that EXD2 KO cells have enhanced ALT phenotypes (TIF formation, RPA colocalization to telomeres, C-circle levels, APB levels, T-SCEs, and telomere synthesis in G2 and M). Additionally, they conclude that EXD2 is a suppressor of RAD52-independent telomere synthesis. The authors go on to establish the genetic dependencies of many of their ALT phenotypes induced by EXD2 loss. Additionally, they examine several synthetic lethal interactions with EXD2 loss (BLM, SLX4, DNA2, POLD3). Overall, their work makes an incremental but important contribution to the field, and I recommend publication.

In their rebuttal the authors addressed the majority of concerns raised in the initial review of this manuscript.

One remaining problem with the work is a contradiction raised regarding telomere length and productive telomere synthesis as outlined below:

EXD2 KO is associated with shorter telomere length which is in contradiction to the claim that EXD2 KO cells have elevated levels of conservative telomere synthesis which is thought to be "productive"

(telomere lengthening).

In their rebuttal the authors claim that these results are not inconsistent because it is possible that RAD52-independent synthesis, which is utilized in EXD2 KO cells is "less productive/efficient than the RAD52-dependent mechanism normally utilized by ALT-reliant cells, resulting in reduced average telomere length."

However, I disagree. What does it mean to have more "productive" synthesis that is also less "productive/efficient"? Either their assay is not truly measuring productive synthesis or there must be some enhancement of telomere attrition in the EXD KO cells which is not being addressed.

I don't think this contradiction should prevent publication, but I think it is important enough that it warrants an explicit acknowledgement/description in the main text of the paper with potential explanations as to its cause.

Manuscript NCOMMS-22-12191-T

We have performed additional experiments requested by the reviewers, that we believe further strengthen the manuscript.

Reviewer #1:

The quality of the data in the revised manuscript by Broderick and colleagues has been dramatically improved.

Our reply: We thank the reviewer for recognising the extent and the quality of the work performed to address the comments/suggestions raised during the first round of review.

The changes shown in Fig 2B and Fig S1C- are quite mild. γ H2AX should be a better DNA damage marker, which can better capture the mixed forms of DNA breaks induced by fork collapse. In addition, the authors should also perform pChk1 staining, which would re-enforce their hypothesis that EXD2 deficiency causes replication problems at the ALT telomeres.

Our reply: As requested by the reviewer we have performed analysis of γ H2AX co-localisation with telomeres in WT and EXD2^{-/-} cells. This new dataset shows an increased level of γ H2AX foci in EXD2-deficient cells at telomeres as compared to the WT control. This is consistent with our previous analysis of other DDR markers such as 53BP1 or RPA. This new data is now incorporated in **New Supplementary Fig, 1d**.

Regarding pCHK1 staining – despite several attempts we were unable to obtain a clear focal distribution of pCHK1, which would be consistent with the fact that CHK1 upon activation (phosphorylation) dissociates from chromatin¹⁻³.

Telomeric SMARD analysis is a more direct method to address whether there is DNA replication defects at telomeres. The authors need to perform the SMARD assay using the EXD2^{-/-} cells.

Our reply: We have shown in multiple cell lines, including ALT-reliant U2OS cells that the lack of EXD2 is associated with a global replication defects utilizing another technology i.e. DNA fibre, which allows for monitoring of individual replication forks in live cells (⁴ and this manuscript) . This is a highly sensitive and widely used approach to monitor replication fork progression. In addition, we have also shown, in the current manuscript, that targeting the replication fork remodeler SMARCAL1 ameliorates DNA damage responses at telomeres associated with the loss of EXD2. Thus, in our view the requested SMARD-based analysis (technology that is a modification of DNA fibre) would not provide additional insight into the regulation of RAD52-dependent and RAD52-independent branches of BIR. In addition, SMARD is a highly specialised technology that is being utilized only by a few laboratories worldwide.

For the C-circle blot in Fig 3A, visually, I am not convinced that there is any difference between U2OS and EXD2^{-/-} cl.1. Similarly, in Fig 3B, the C-circle increases in siEXD2 U2OS cells is also very mild. To confirm these very mild C-circle increases in the EXD2 deficient U2OS cells, both EXD2^{-/-} cl.1 and siEXD2 need to be rescued.

Our reply: As requested by the reviewer we have performed this analysis. The new data show that only the WT EXD2 but not the nuclease-dead version of the protein is capable of complementing the defect associated with the C-circles (Figure 1, below). We would also like to stress, that this new data is consistent with the initial analysis requested by the reviewer i.e., “to perform additional experiments with the EXD2 KO cell lines complemented with either, WT or EXD2 -nuclease-dead version of the protein” – which we have performed. All these

analysis showed that only the wild-type but not the nuclease-dead version of the protein is able to rescue the phenotype associated with EXD2 loss (**Fig. 2c and Supplementary Fig. 3c**).

Figure 1: a) Quantification of C-circles in WT U2OS, EXD2^{-/-} and pools of EXD2^{-/-} cells recomplemented with either WT or nuclease dead (ND) FLAG-EXD2 (n=3 independent experiments, statistical significance determined by student's t-test, bars represent +/- SEM).

In addition, analysis of C-circles was performed not only in EXD2 KO cells but also in a panel of ALT-reliant cells (including a revertant control cell line that no longer relies on ALT mechanism). In all instances we employed densitometry analysis with appropriate controls in place and detected a clear and highly statistically significant difference between the WT cells and cells deficient or depleted for EXD2 (**Fig. 3b**; 7 independent cell lines in total, including U2OS EXD2 KO). Moreover, the significance of differences between WT and EXD2-deficient cells for various phenotypes are determined by appropriate statistical analyses utilising multiple biological repeats.

Minor points:

- Line-109: mis-spelled "Dysfunction"

Our reply: This has been corrected in the new version of the manuscript.

- Please provide a reference for the validity of TeloSizer

Our reply: We have now included the requested reference.

Reviewer #3:

One remaining problem with the work is a contradiction raised regarding telomere length and productive telomere synthesis as outlined below:

EXD2 KO is associated with shorter telomere length which is in contradiction to the claim that EXD2 KO cells have elevated levels of conservative telomere synthesis which is thought to be "productive" (telomere lengthening).

In their rebuttal the authors claim that these results are not inconsistent because it is possible that RAD52-independent synthesis, which is utilized in EXD2 KO cells is "less

productive/efficient than the RAD52-dependent mechanism normally utilized by ALT-reliant cells, resulting in reduced average telomere length."

However, I disagree. What does it mean to have more "productive" synthesis that is also less "productive/efficient"? Either their assay is not truly measuring productive synthesis or there must be some enhancement of telomere attrition in the EXD KO cells which is not being addressed.

Our reply: We agree that the elevated number of EdU incorporation events and conservative synthesis events seen in EXD2-deficient cells compared to WT vs the average telomere length observed in these cells could be seen as an apparent contradiction.

However, the assays carried out measure the frequency of incorporation events, and not their average length- so we do think it is possible that RAD52-independent BIR may operate with a slower kinetics (i.e., the average length of DNA synthesis is shorter than in RAD52-dependent BIR). The reviewer, however, raises an important point, i.e., that there could be a telomere attrition mechanism occurring in EXD2-deficient cells leading to shorter average telomere length in these cells (where the processivity of RAD52-independent BIR is similar to that in RAD52-dependent BIR). Therefore, as requested, we have included a specific discussion of the possible explanations (in our view) for this observation in the revised manuscript.

Briefly, we can envision two possible explanations for our observations. Firstly, as RAD52-independent BIR has been shown to require a different annealing factor (possibly RAD51AP1 according to a recently published paper by the Zou lab⁵ and we show that the RAD52-independent BIR carried out in EXD2-deficient cells requires the action of the MUS81 nuclease and SMARCAL1, we consider that MUS81-dependent cleavage of regressed stalled replication forks can act to initiate RAD52 independent BIR, which operates via a different annealing factor than RAD52-dependent BIR. As stated above, the assays used in our study measure the frequency of EdU incorporation at the telomere, and conservative DNA synthesis at telomeres but not the average length of synthesis. We therefore consider it possible that BIR initiated by MUS81-dependent cleavage of regressed forks (as opposed to XPF-mediated DNA cleavage as likely utilised by RAD52-dependent BIR⁶) may on average lead to shorter tracts of DNA synthesis. Moreover, as we discuss in the original version of the manuscript, work from the Ira lab in yeast⁷ shows that BIR can operate by multiple rounds of engagement, whereby migrating D-loops formed during BIR-mediated synthesis can be cleaved by Mus81, with the generated single-ended DNA break used to initiate strand invasion and re-initiate synthesis. In EXD2-deficient cells we show that MUS81 is required for the elevated EdU incorporation at telomeres and conservative synthesis as assayed by two-round CO-FISH. We consider it possible that MUS81 could function early (cleavage of regressed forks) and later during this process (cleavage of the migrating D-loops formed during conservative DNA synthesis) leading to the generation of a new single-ended break, which itself could be used to re-engage RAD52-independent BIR in EXD2 deficient cells (with this process potentially occurring multiple times). Therefore, if the efficiency of the annealing factor utilised in RAD52-independent BIR, or the propensity for MUS81-dependent cleavage of migrating D-loops in RAD52-independent BIR differs to that observed in RAD52-dependent BIR, this could explain the increased frequency of EdU incorporation events and conservative synthesis events observed in EXD2-deficient cells, while the average telomere length is shorter than that observed in WT cells. Moreover, it may be possible that RAD52-dependent and -independent BIR may favour different templates, i.e., may have different preferences for intra- vs inter-chromosomal strand annealing which could conceivably lead to differences in average telomere length.

Secondly, as pointed out by the reviewer, it is possible that the efficiency of RAD52-independent BIR is similar or even greater than the RAD52-dependent mechanism and that the reason for the shorter average telomere length observed in EXD2-deficient cells is due to a telomere attrition mechanism, for example the degradation of nascent telomeric DNA synthesised in EXD2-deficient cells. Since EXD2-deficient cells show an increased nucleolytic

degradation of nascent DNA due to an inability to counteract excessive replication fork reversal⁴ it is possible that excessive nucleolytic degradation contributes to telomere attrition (i.e that degradation of telomeric sequences may occur before the engagement of BIR).

- 1 Smits, V. A., Reaper, P. M. & Jackson, S. P. Rapid PIKK-dependent release of Chk1 from chromatin promotes the DNA-damage checkpoint response. *Current biology : CB* **16**, 150-159 (2006). <https://doi.org:10.1016/j.cub.2005.11.066>
- 2 Halder, S. *et al.* SPRTN protease and checkpoint kinase 1 cross-activation loop safeguards DNA replication. *Nature communications* **10**, 3142 (2019). <https://doi.org:10.1038/s41467-019-11095-y>
- 3 Lukas, C., Falck, J., Bartkova, J., Bartek, J. & Lukas, J. Distinct spatiotemporal dynamics of mammalian checkpoint regulators induced by DNA damage. *Nature cell biology* **5**, 255-260 (2003). <https://doi.org:10.1038/ncb945>
- 4 Nieminuszczy, J. *et al.* EXD2 Protects Stressed Replication Forks and Is Required for Cell Viability in the Absence of BRCA1/2. *Molecular cell* **75**, 605-619.e606 (2019). <https://doi.org:10.1016/j.molcel.2019.05.026>
- 5 Yadav, T. *et al.* TERRA and RAD51AP1 promote alternative lengthening of telomeres through an R- to D-loop switch. *Molecular cell* **82**, 3985-4000.e3984 (2022). <https://doi.org:10.1016/j.molcel.2022.09.026>
- 6 Guh, C.-Y. *et al.* XPF activates break-induced telomere synthesis. *Nature communications* **13**, 5781 (2022). <https://doi.org:10.1038/s41467-022-33428-0>
- 7 Mayle, R. *et al.* DNA REPAIR. Mus81 and converging forks limit the mutagenicity of replication fork breakage. *Science (New York, N.Y.)* **349**, 742-747 (2015). <https://doi.org:10.1126/science.aaa8391>